# From Holo Pockets to Electron Density: GPT-style Drug Design with Density

**Jiahao Chen** [* 1 2 3 4] **Letian Gao** [* 5] **Yanhao Zhu** [* 4] **Wenbiao Zhou** [* 5] **Bing Su** [1 2 3] **Zhi John Lu** [5] **Bo Huang** [4 6]

## Abstract

Recent advances in generative modeling have enabled significant progress in structure-based drug design (SBDD). Existing methods typically condition molecule generation on empty binding pockets from holo complexes, overlooking informative components such as the filler (ligands and solvent). Here, we leverage low-resolution electron density (ED) derived from the filler as a physically grounded condition for *de novo* drug design. We consider two types of ED—calculated and cryo-EM/X-ray—obtainable from computational or experimental sources, supporting unified pretraining and experimental integration. Compared with rigid pocket representations, experimental ED naturally captures conformational flexibility and provides a more faithful description of the binding environment. Based on this, we introduce EDMolGPT, a decoder-only autoregressive framework that generates molecules from low-resolution ED point clouds. By grounding generation in physically meaningful density signals, EDMolGPT mitigates structural bias and produces molecules with 3D conformations. Evaluations on 101 biological targets verify the effectiveness. Our project page: https://jiahaochen1.github.io/EDMolGPT_Page/.

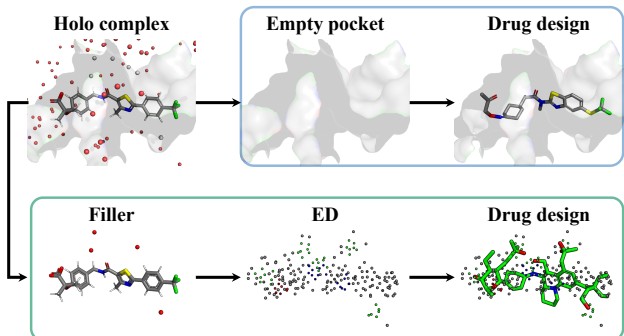

*Figure 1.* Comparison between pocket-based drug design (blue-circled region) and our electron density (ED)-based drug design framework (green-circled region). The red dots denote the solvent. Filler is defined as all elements within a $4.5\text{Å}$ radius of the ligand, excluding the binding pocket.

## 1. Introduction

AI-driven drug design has emerged as a powerful paradigm for generating molecules that selectively bind biological targets. Among various strategies, structure-based drug design (SBDD) has attracted significant attention, as it conditions molecular generation on the three-dimensional geometry of a binding site. As shown in Fig. 1, most existing SBDD pipelines begin from a holo protein–ligand complex and remove the filler (e.g., ligands and solvent molecules) to construct an explicit binding pocket, which is then treated as a fixed scaffold for ligand generation or optimization. This formulation implicitly assumes that the binding pocket can be accurately delineated and represented by a single static conformation, an assumption that suppresses intrinsic protein flexibility and fails to capture conformational adaptations associated with ligand binding.

To address this limitation, many approaches in related fields have sought to account for protein flexibility. For example, pocket ensemble-based methods (Szabó et al., 2021) partially address this limitation for molecular docking, but they are difficult to integrate into molecular generation frameworks, which typically require a unified and fixed conditioning representation rather than a collection of discrete conformations. Experimental electron density (ED) offers a promising alternative, providing a continuous, physics-grounded representation that encodes ensemble-averaged spatial distributions, physicochemical environments, and interaction patterns (Ding et al., 2022b; Ma et al., 2023),

---
[*]Equal contribution [1]Gaoling School of Artificial Intelligence, Renmin University of China, Beijing, China. [2]Beijing Key Laboratory of Research on Large Models and Intelligent Governance, Beijing, China. [3]Engineering Research Center of Next Generation Intelligent Search and Recommendation, MOE, Beijing, China. [4]NeoPrimeTech Biology, Beijing, China [5]MOE Key Laboratory of Bioinformatics, State Key Laboratory of Green Biomanufacturing, Center for Synthetic and Systems Biology, School of Life Sciences, Tsinghua University, Beijing, China. [6]College of Pharmaceutical Sciences, Capital Medical University, Beijing, China.. Correspondence to: Bing Su <subingats@gmail.com>, Zhi John Lu <zhilu@tsinghua.edu.cn>, Bo Huang <bohuang_011@163.com>.

*Proceedings of the 43$^{rd}$ International Conference on Machine Learning*, Seoul, South Korea. PMLR 306, 2026. Copyright 2026 by the author(s).

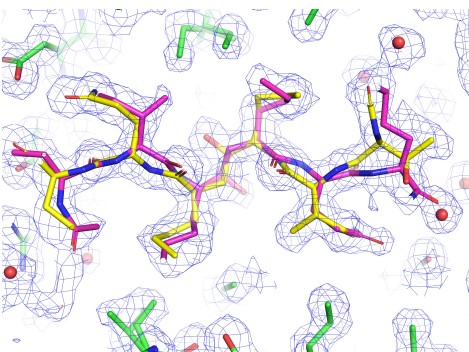

*Figure 2.* Experimental ED reflects conformational dynamics of a filler in a protein pocket (PDB ID: 6KMP). The experimental ED map is shown as blue mesh, representing the ensemble-averaged electron density derived from X-ray diffraction. Protein atoms are shown as green sticks. The ligand is shown in yellow and purple sticks, with colors corresponding to alternative conformations resolved in the density, indicative of conformational dynamics in the bound state. Water molecules are shown as red spheres. The electron density profile of the filler, including the ligand and proximal solvent molecules, is indicated by the orange dashed line.

thereby avoiding reliance on rigid geometric abstractions.

While recent studies (Wang et al., 2022) have explored molecular generation using experimental ED of binding pockets, in practice pocket ED is frequently weak or poorly resolved in highly flexible regions, precisely where conformational variability is most pronounced, leading to unstable or ambiguous conditioning signals for learning-based models. In contrast, filler ED is typically well-defined, experimentally validated, and spatially localized, providing a more reliable and informative conditioning signal for generative modeling. As an experimentally grounded, continuous representation, filler ED encodes ensemble-averaged spatial distributions and interaction patterns, enabling conformational variability to be captured without reliance on rigid geometric assumptions or hand-crafted abstractions, distinguishing it from heuristic soft representations such as 3D pharmacophores or interaction fields. These considerations motivate an underexplored question: can filler ED serve as a flexible and experimentally grounded conditioning representation for molecular generation in drug discovery?

While ED provides physically grounded input for drug design, extracting ED from the filler and constructing model-friendly representations remains underexplored. Unlike previous methods (Li et al., 2025; Zhang et al., 2024) that rely on ED from rigid, empty pockets, a holo complex inherently contains filler components (ligands and solvent) that encode the conformational flexibility and interaction patterns of the binding site. We directly derive ED representations from the filler, bypassing the intermediate step of modeling the pocket. Specifically, we consider two types of ED: (1) **calculated electron density (CalED)**, derived analytically from atomic coordinates using physical scattering models for efficient pre-training, and (2) **cryo-EM/X-ray derived**

density (**ExpED**), obtained from experimental reconstructions. As shown in Fig. 2, ExpED captures measurement noise, conformational flexibility, and all filler interactions, providing a comprehensive view of the binding environment. Leveraging ExpED enables the model to generate ligands compatible with the dynamic pocket, yielding realistic conformations and diverse scaffolds. ED point clouds are sampled from the density maps and annotated with pharmacophore features, offering chemically meaningful guidance. By combining CalED and ExpED, we can unifies scalable pre-training with experimental signals, capturing flexibility and all-pocket interactions for accurate ligand generation.

For the algorithm, we propose EDMolGPT, a decoder-only autoregressive framework for 3D drug design conditioned on low-resolution ED represented as a point cloud. To account for the importance of input order in GPT-style models, we reorder the point cloud by spatial coordinates. Generated molecules are represented using FSMILES (Feng et al., 2024), which captures rational molecular conformations. Unlike most existing ligand generation frameworks that rely on encoder–decoder architectures or diffusion models, EDMolGPT is the first decoder-only approach, combining simplicity, flexibility, and high efficiency while fully leveraging model capacity to produce accurate 3D structures. The model is pre-trained on large-scale calculated electron density from public datasets and fine-tuned on cryo-EM/X-ray ligand densities from experimental measurements, which are more realistic but limited in quantity. At test time, molecules are generated conditioned on the ED of filler components, as shown in Fig. 4. Experiments on the DUD-E dataset verify that EDMolGPT produces molecules with conformations compatible with the binding pocket and bioactivity. Our contributions can be summarized as follows:

(1) Instead of conditioning on empty pockets, we generate molecules directly from the ED derived from the filler. We consider both CalED and ExpED, enabling unified large-scale pre-training and experimental integration. **To the best of our knowledge, this is the first work to incorporate the filler's cryo-EM/X-ray derived density into generative modeling for structure-based drug design.**

(2) We introduce EDMolGPT, a decoder-only autoregressive model for 3D drug design that is conditioned on low-resolution ED representing the binding environment. This approach addresses the limitations of rigid pocket representations, allowing for the generation of molecules compatible with the dynamic nature of protein binding sites.

(3) Through extensive experiments on up to 101 targets from DUD-E dataset, EDMolGPT consistently generates molecules with both favorable 3D conformations compatible with the target binding pocket and demonstrated bioactivity, validating its potential for *de novo* drug discovery.

## 2. Related work

**Structure-based drug design** SBDD generates ligands by exploiting the 3D structure of a target receptor. Classical SBDD workflows, such as molecular docking (Morris et al., 2009), scoring functions (Breda et al., 2008), and molecular dynamics (MD) simulations (Hollingsworth & Dror, 2018), are computationally expensive, particularly for large-scale virtual screening. To address these limitations, recent advances integrate AI-based generative modeling, with progress in both autoregressive (Gao et al., 2022) and diffusion-based approaches (Xu et al., 2022). Among autoregressive methods, Pocket2Mol (Peng et al., 2022) introduced an E(3)-equivariant generative framework that samples valid molecules from pocket geometry, improving affinity and diversity. Lingo3DMol (Feng et al., 2024) further incorporated fragment-based SMILES with 3D geometric features to enable language-model-driven molecule generation. In diffusion-based methods, TargetDiff (Guan et al., 2023a) conditions on protein pocket information to generate ligands with high binding affinity, while MolCRAFT (Qu et al., 2024) performs noise-controlled sampling for stable conformations and superior docking scores. Different from them, our EDMolGPT generates full 3D ligand conformations conditioned on point clouds extracted from the filler's low-resolution electron density, enabling the generation of novel valid molecules with accurate structural geometry.

**Electron density-guided molecule generation** Recent advances have incorporated electron density (ED) into AI-driven molecule generation, yet existing methods struggle to balance scaffold novelty, 3D conformation fidelity, and drug-likeness under binding constraints. Wang et al. (Wang et al., 2022) introduced the first ED-guided generative framework using a two-stage pipeline that predicts ligand densities and subsequently assembles molecules via fragments, which may propagate errors and limit scaffold diversity. ED2Mol (Li et al., 2025) treats ED as an auxiliary constraint for fragment-based assembly, improving chemical plausibility but still restricting scaffold exploration. ECloud-Gen (Zhang et al., 2024) conditions sequence generation on ED for *de novo* design, yet lacks explicit 3D reasoning, potentially compromising binding conformations. In contrast, our method directly exploits low-resolution ED as a continuous 3D field to guide end-to-end atomic placement, enabling novel scaffold generation with accurate geometry, strong binding compatibility, and favorable drug-likeness.

## 3. Method

In Sec. 3.1, we formulate the problem of electron density-based drug design. Building upon this, Sec. 3.2 describes the extraction of point clouds from electron density. In Sec. 3.3, we present the representation of molecular struc-

tures, including FSMILES and relative distances. Finally, Sec. 3.4 details the overall EDMolGPT architecture and the procedures for training and inference, specifically how to generate a molecule conditioned on a given point cloud.

### 3.1. Problem formulation

The goal of drug design is to generate a molecule $\mathcal{M} = \{(a_m^i, \boldsymbol{v}_m^i)\}_{i=1}^{N_m}$, consisting of $N_m$ atoms, where $a_m^i \in \mathbb{R}^1$ denotes the atom type and $\boldsymbol{v}_m^i \in \mathbb{R}^3$ represents its position in 3D space, with three components corresponding to the $x$, $y$, and $z$ coordinates. Our method conditions on point clouds extracted from the filler $\mathcal{F}$ of complexes. Specifically, we construct a compact point cloud representation $\mathcal{P}_f = \{(c_p^i, \boldsymbol{v}_p^i)\}_{i=1}^{N_p}$ from $\mathcal{F}$, where $c_p^i$, $\boldsymbol{v}_p^i$, and $N_p$ denote the point types, coordinates, and number of points, respectively. This geometric representation provides a rich yet compact conditioning signal, enabling the model to capture the binding context sufficiently. The specific procedures for obtaining the filler $\mathcal{F}$, as well as the differences between training and inference, are detailed in Sec. 3.4.

### 3.2. Generating point cloud

The primary distinction between CalED and ExpED lies in their data sources: CalED is derived from solved structures in real space, whereas ExpED is obtained directly from raw experimental observations in reciprocal space. Accordingly, CalED is generated by first transforming the solved structures into reciprocal space via Fast Fourier Transform (FFT), while ExpED requires no such transformation step.

For CalED, given a filler structure, we apply the FFT to compute its electron diffraction pattern. Let $\{\boldsymbol{v}_f^i\}_{i=1}^{N_f}$ denote the atomic coordinates of filler $\mathcal{F}$. The corresponding structure factors in reciprocal space are computed as

$$F(\boldsymbol{h}) = \sum_{i=1}^{N_f} f_i(\boldsymbol{h}) \, e^{2\pi i \boldsymbol{h} \cdot \boldsymbol{v}_f^i}, \qquad (1)$$

where $\boldsymbol{h}$ is the reciprocal-lattice vector and $f_i(\boldsymbol{h})$ denotes the atomic scattering factor of atom $i$. In contrast, these computational steps are unnecessary for ExpED, since it can be directly derived from experimental measurements.

To further control the spatial resolution of ED, we apply a high-frequency cutoff based on the minimum interplanar spacing $d_{\min}$, such that only spatial frequencies corresponding to features larger than $d_{\min}$ are retained. The filtered diffraction data are subsequently transformed back into real space to reconstruct a smooth electron density map, from which 3D point clouds are sampled (Fig. 3). The ED is obtained via truncated inverse Fourier transformation:

$$\rho(\boldsymbol{v}_f) = \frac{1}{V} \sum_{|\boldsymbol{h}| \leq 1/d_{\min}} F(\boldsymbol{h}) \, e^{-2\pi i \boldsymbol{h} \cdot \boldsymbol{v}_f}. \qquad (2)$$

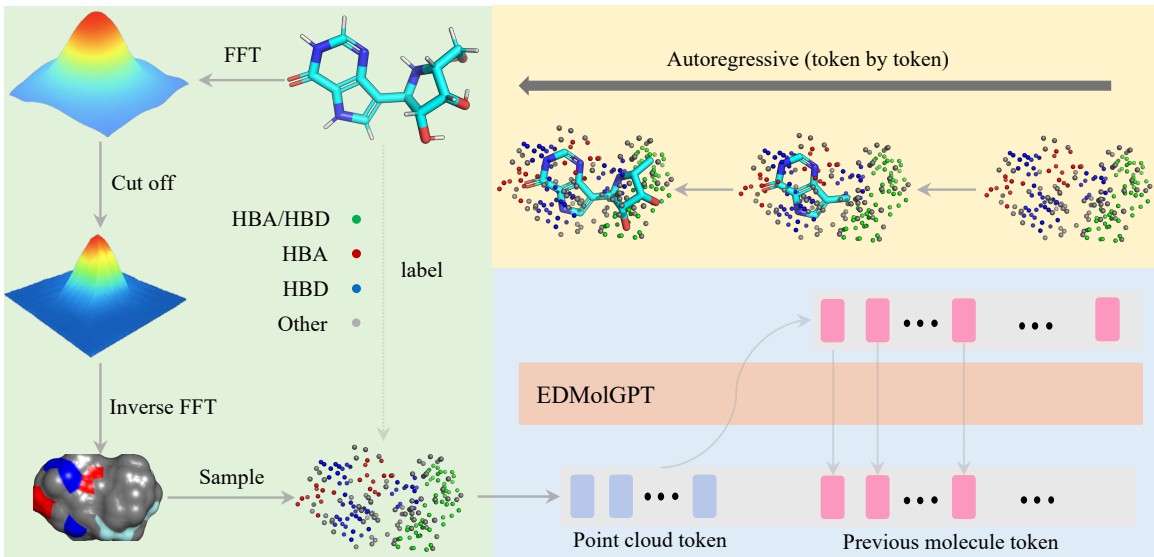

*Figure 3.* The overall pipeline of our method. The components shown with a green background correspond to the generation of 3D point clouds from the input ligand. The blue-highlighted components represent the molecule generation process, where each molecular token is predicted sequentially based on the point cloud tokens and the previously generated molecular tokens. Finally, the steps highlighted in yellow illustrate how the predicted molecular structure progressively occupies and fills the sampled point clouds, providing an interpretable view of the generation process.

We then randomly sample $N_p$ points from $\rho(\boldsymbol{v}_f)$ to generate a set of low-resolution ED point clouds $\{\boldsymbol{v}_p^i\}_{i=1}^{N_p}$. While these point clouds capture the overall filler structure of the ligand, they contain limited pharmacophore information, posing challenges for the generation of bioactive molecules. To enrich the chemical features, for each point in the cloud, we compute its minimal distance to all atoms in the filler $\mathcal{F}$ and assign a pharmacophore type based on the closest atom. Specifically, each point is assigned a type indicator $c_p^i$, whose value is selected from {hydrogen bond donor (HBD), hydrogen bond acceptor (HBA), hydrogen bond donor/acceptor (HBD / HBA), Other }. Finally, we obtain a set of labeled point clouds $\mathcal{P}_f = \{(c_p^i, \boldsymbol{v}_p^i)\}_{i=1}^{N_p}$. Since autoregressive models are sensitive to input ordering, we sort the points in $\mathcal{P}_f$ in ascending order of their $x$, $y$, and $z$ coordinates, thereby providing consistent input.

### 3.3. Input format of molecule

Determining the ordering of the molecular structure $\mathcal{M}$ is crucial for autoregressive modeling. A straightforward approach is to use SMILES (Weininger, 1988) to represent the molecule together with its absolute spatial positions. While this representation is sufficient to describe molecular structures, its application in autoregressive generation often results in unrealistic or physically inconsistent conformations (Feng et al., 2024; Qu et al., 2024). To overcome this limitation, we adopt a modified Lingo3DMol representation (Feng et al., 2024) for $\mathcal{M}$, yielding $\widehat{\mathcal{M}} = \{(\widehat{a}_m^i, \widehat{\boldsymbol{v}}_m^i, \widehat{l}_m^i, \widehat{\theta}_m^i, \widehat{\phi}_m^i)\}$, where $\widehat{a}_m^i, \widehat{\boldsymbol{v}}_m^i, \widehat{l}_m^i, \widehat{\theta}_m^i,$

and $\widehat{\phi}_m^i$ denote the Fragment SMILES (FSMILES) token, discretized 3D coordinates, bond length, bond angle, and dihedral angle, respectively. In the following section, we detail the procedure for converting a given molecule $\mathcal{M}$ into its representation $\widehat{\mathcal{M}}$.

**FSMILES** FSMILES (Feng et al., 2024) is a novel 2D molecular representation derived from SMILES, which decomposes molecules into fragments while retaining the standard SMILES syntax for each fragment. Compared with SMILES, FSMILES improves the learning of 2D molecular patterns by representing fragments and local structures with dedicated symbols and by prioritizing ring closures, which facilitates the generation of molecules with correct ring structures and bond angles. However, in the original FSMILES, edges connecting atoms within a ring were often cut, which could lead to overly fragmented molecular representations. Therefore, we improve FSMILES, avoiding splitting small fragments that link rings, thereby reducing excessive fragmentation and preserving more of the molecule's structural integrity. More details are in Appendix Sec. C.1.

**Discretized 3D coordinates** The coordinates of a molecule $\mathcal{M}$ are originally continuous in three-dimensional space. To make them compatible with autoregressive modeling, we discretize the spatial coordinates following the input format of Lingo3DMol. Specifically, we first compute the geometric center of $\mathcal{M}$, denoted as $\boldsymbol{\mu}_m \in \mathbb{R}^3$, and obtain the initial discretized coordinates $\widetilde{\boldsymbol{v}}_m^i = \left\lfloor \frac{\boldsymbol{v}_m^i - \boldsymbol{\mu}_m}{\sigma} \right\rfloor$, where $\sigma$ and $\lfloor \cdot \rfloor$ denote a scaling factor and rounding operation, respectively. Since the spatial extent of most drug-like

molecules lies within 5–30 Å, we set $\sigma = 0.1$, mapping the coordinates into a bounded integer grid of moderate resolution (within $[-150, 150]$ along each axis). To facilitate autoregressive prediction, we shift all discretized coordinates by a constant offset so they become positive integers. The final coordinates, denoted as $\widehat{v}_m^i$, preserve geometric detail while keeping the vocabulary size manageable, thereby improving tractability and training stability. Point clouds are also transformed into this shifted space, denoted as $\{\widehat{v}_p^i\}$. More details are in Appendix Sec. C.2.

**Relative Distance**   Although the discretized coordinates $v_m^i$ capture the absolute spatial positions of atoms, we further incorporate relative geometric information to explicitly model local structural dependencies, which is beneficial for autoregressive inference. Specifically, for each atom $v_m^i$, we consider its three preceding atoms $v_m^{i-1}$, $v_m^{i-2}$, and $v_m^{i-3}$, and compute the the bond length $l_m^i$, bond angle $\theta_m^i$, and dihedral angle $\phi_m^i$ as follows:

$$l_m^i = \left\| v_m^i - v_m^{i-1} \right\|_2, \tag{3}$$

$$\theta_m^i = \arccos\left( \frac{\left( v_m^{i-1} - v_m^{i-2} \right) \cdot \left( v_m^i - v_m^{i-1} \right)}{\left\| v_m^{i-1} - v_m^{i-2} \right\|_2 \left\| v_m^i - v_m^{i-1} \right\|_2} \right), \tag{4}$$

$$\phi_m^i = \arctan\Big( \big( (b_1 \times b_2) \times (b_2 \times b_3) \big)$$
$$\cdot \frac{b_2}{\|b_2\|_2}, (b_1 \times b_2) \cdot (b_2 \times b_3) \Big), \tag{5}$$

where $b_1 = v_m^{i-2} - v_m^{i-3}$, $b_2 = v_m^{i-1} - v_m^{i-2}$, and $b_3 = v_m^i - v_m^{i-1}$. We similarly convert $l_m^i$, $\theta_m^i$, and $\phi_m^i$ into discrete representations for autoregressive modeling:

$$\widehat{l}_m^i = \left\lfloor \frac{l_m^i}{\sigma} \right\rfloor, \quad \widehat{\theta}_m^i = \left\lfloor \frac{\theta_m^i}{10} \right\rfloor, \quad \widehat{\phi}_m^i = \left\lfloor \frac{\phi_m^i}{10} \right\rfloor \tag{6}$$

As shown in Eq. 6, we discretize bond lengths using the same rule as coordinates. For bond angles and dihedral angles, we apply a coarser discretization, dividing the 180-degree range into 10-degree intervals. This design ensures that relative geometric information contributes effectively while preserving the learnability of the task. More details are in the Appendix Sec. C.3

### 3.4. EDMolGPT

**Training**   The overall architecture of EDMolGPT follows GPT-2 (Radford et al., 2019), a decoder-only framework and adopt the default Transformer positional embeddings as used in GPT. Positional embeddings provide ordering and global spatial context, improving spatial awareness during molecular decoding. We remove all components in the filler except the ligand, and use the resulting ED as the conditioning signal. As a distinction, we denote the ED used

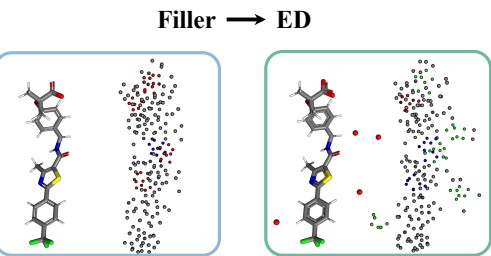

Filler ⟶ ED

*Figure 4.* Difference between training (blue) and inference (green): ED during training is derived from the ligand, while ED during inference incorporates solvent(red dots).

for training as $\widehat{\mathcal{P}}_m$. During training, we concatenate the point cloud and molecule sequences and feed them into ED-MolGPT to predict the molecule token-by-token. Formally, after acquiring the discretized point cloud $\widehat{\mathcal{P}}_m$ and the corresponding molecule $\widehat{\mathcal{M}}$, the input features for the point cloud and molecule are defined as:

$$\begin{aligned} h_p^i &= g_x(\widehat{v}_{p,x}^i) + g_y(\widehat{v}_{p,y}^i) + g_z(\widehat{v}_{p,z}^i) + g_c(c_p^i), \\ h_m^i &= g_x(\widehat{v}_{m,x}^i) + g_y(\widehat{v}_{m,y}^i) + g_z(\widehat{v}_{m,z}^i) + g_a(\widehat{a}_m^i), \end{aligned} \tag{7}$$

where $g_x$, $g_y$, $g_z$, $g_c$, and $g_a$ denote the embedding functions for X-, Y-, and Z-coordinates, point cloud type, and FSMILES token, respectively. Note that the coordinate embedding functions are shared between the point cloud and molecule, as they reside in the same spatial space. Since all variables are discretized into categorical spaces, the model predicts $\widehat{a}_m^t$, $\widehat{v}_m^t$, $\widehat{l}_m^t$, $\widehat{\theta}_m^t$, and $\widehat{\phi}_m^t$ using linear classification heads followed by softmax normalization, and is optimized with the CE loss. The overall training objective is defined as:

$$\mathcal{L}_{\text{EDMolGPT}} = -\frac{1}{N_m} \sum_{t=1}^{N_m} \log p\Big( (\widehat{a}_m^t, \widehat{v}_m^t, \widehat{l}_m^t, \widehat{\theta}_m^t, \widehat{\phi}_m^t) \Big|$$
$$\underbrace{h_p^1, \ldots, h_p^{N_p}}_{\text{point cloud features}}, \quad \underbrace{h_m^1, \ldots, h_m^{t-1}}_{\text{previous molecule features}} \Big). \tag{8}$$

**Inference**   During inference, we feed the filler's ED (It contains the information of solvent) into EDMolGPT and generate the molecular sequence $\mathcal{M}$ in an autoregressive, token-by-token manner. For FSMILES tokens $\widehat{a}_m^i$ and relative geometric tokens $\widehat{l}_m^i, \widehat{\theta}_m^i, \widehat{\phi}_m^i$, we apply temperature sampling (Radford et al., 2019) to draw predictions from the model's output distribution. Instead of directly sampling discretized 3D coordinates, we exploit the predicted relative geometric features to restrict the sampling space for $v_m^i$. Specifically, given the three previously generated atom positions $v_m^{i-1}$, $v_m^{i-2}$, $v_m^{i-3}$, and predicted $(l_m^i, \theta_m^i, \phi_m^i)$, we recover the continuous bond length, bond angle, and dihedral angle from their discretized representations. These quantities uniquely define a local reference frame, within

which the feasible set of $\boldsymbol{v}_m^i$ lies on a spherical surface parameterized by $(l_m^i, \theta_m^i, \phi_m^i)$. The model then samples $\boldsymbol{v}_m^i$ from this constrained space, ensuring geometric consistency with the previously generated atoms while reducing the search space and improving the stability of autoregressive inference. More details about how to apply relative distance during inference are in Appendix Sec. D

# 4. Experiment

## 4.1. Settings

**Datasets** Our EDMolGPT model is pre-trained on publicly available datasets[1], which include approximately eight million molecules. To improve data quality, we further filter the dataset using the Quantitative Estimate of Drug-likeness (QED) and the Synthetic Accessibility Score (SAS), resulting in a curated set of approximately two million molecules. We leverage a large-scale molecular dataset to generate CalED for pre-training. For fine-tuning, we collect complex data from 40k binding experiments in PDBbind and construct ExpED accordingly. For point cloud generation, we set $d_{\min} = 3.5$Å for each molecule. All point clouds are standardized to contain $N_p = 199$ points, which we find to offer a favorable trade-off between performance and computational efficiency. For evaluation, we adopt DUD-E (Mysinger et al., 2012) dataset, which contains 101 receptors and their corresponding binding molecules.[2] We make the CalED from the ligand, and for ExpED, we construct the filler density by including the ligand itself together with all solvent and water within a radius of $4.5$Å centered at the ligand. Due to the limited availability of experimental cryo-EM/X-ray density maps, only 92 structures in DUD-E were matched with corresponding experimental densities, and all evaluations on ExpED are therefore conducted on this subset. For each receptor, we generate 1000 molecules for inference. More details about distributions between the training data and inference data in Appendix Sec. F.1.

**Training/Evaluation Details** EDMolGPT is trained following the general setup of GPT-medium with a 24-layer Transformer backbone. We optimize the model using the AdamW (Loshchilov & Hutter, 2017) optimizer with a learning rate of $1 \times 10^{-5}$ and apply a warm-up schedule for the first 1000 steps before decaying according to a cosine schedule . The training is conducted with a batch size of 96 for 100 epochs. All experiments are performed on two NVIDIA A40 GPUs. For inference, we set the temperature $T = 0.7$

---

[1]Data available at: http://data.aicnic.cn/dms-html/dataset_detail.html?id=848

[2]Traditional DUD-E contains 102 targets. But following previous works (Feng et al., 2024), the target with the PDB ID 2H7L in the DUD-E dataset was excluded as it is listed as an obsolete entry in the PDB.

to scale the sampling distribution and make token prediction. We evaluate the pre-trained model on CalED and the fine-tuned model on ExpED, respectively. More details are in Appendix Sec. E.2.

**Evaluation Metrics** We compare EDMolGPT with previous methods utilizing ED: ECloudGen (Zhang et al., 2024) and ED2Mol (Li et al., 2025). For a fair comparison, we use the CalED derived from pre-existing binder to conduct experiments on ECloudGen, denoted as ECloudGen†. For reference, we report results on experimentally validated bioactive ligands, denoted as Reference. We also compare our method with SBDD methods: Pocket2Mol (Peng et al., 2022), TargetDiff (Guan et al., 2023b), Lingo3DMol (Feng et al., 2024), and MolCRAFT (Qu et al., 2024). We evaluate all methods from four complementary perspectives:

**(1)** Bioactive Molecule Recovery: The percentage of targets for which the generated compounds are similar to known active compounds, as measured by the Tanimoto similarity (Bajusz et al., 2015) of ECFP4 fingerprints (Rogers & Hahn, 2010). If at least one molecule generated by the method satisfies ECFP_TS $> 0.5$, we consider the corresponding active compound for that receptor as recovered. We report the overall recovery rate across all pockets.

**(2)** Binding Affinity: We evaluate the binding affinity of generated ligands using GlideSP (Friesner et al., 2004) under two protocols. In *min-in-place*, the generated conformation is minimized within the original binding pocket while preserving its pose, whereas *redocking* allows fully flexible docking and ligand repositioning. Lower scores indicate stronger predicted binding. To assess pose alignment, we additionally report the fraction of cases where *min-in-place* outperforms *redocking*, suggesting that the generated conformations are already close to favorable binding modes. Glide is adopted due to its wide use, strong enrichment capability, and its role as a standard baseline in scoring function evaluations (Su et al., 2018; Shen et al., 2021).

**(3)** Conformational Stability: Assessed using the strain energy distribution of generated conformers. We evaluate via the commonly used Posecheck (Harris et al., 2023). We report the 25%, 50%, and 75% quantiles to reflect geometric reliability of the molecules.

**(4)** Molecular Properties: Evaluated using multiple criteria: QED ↑, SAS ↓, and average molecular weight. These metrics together capture the practicality and overall quality of the generated molecules. Considering that observing one indicator alone is not very meaningful, we analyze the three indicators together in the following experiments.

## 4.2. Results on CalED

**Results on Bioactive Molecule Recovery**   DUD-E, the dataset used for model evaluation, critically includes over 200 experimentally validated active ligands with measured affinities per target. This feature enables a direct comparison of AI-generated molecules against known active compounds, thereby mitigating the limitations of relying solely on purely computational assessments. Consequently, the Bioactive Molecule Recovery metric becomes an important metric for evaluating molecule generation models (Liu et al., 2024). On this crucial metric (Tab. 1), EDMolGPT achieves the highest recovery ratio among other methods, successfully reproducing bioactive molecules for 41% of the targets. This result implies that nearly half of the targets can be matched with generated compounds that are structurally similar (ECFP4_TS > 0.5) to known actives, highlighting the practical relevance of our method. The high recovery rate indicates that molecules generated by EDMolGPT are not only computationally favorable but also exhibit real biological activity.

**Results on Binding Affinity** As shown in Tab. 1, ED-MolGPT achieves the lowest average Glide score in the min-in-place setting ($-6.92$), indicating that EDMolGPT-generated molecules adopt more favorable conformations for pocket-binding than those produced by other models. Beyond this absolute measure of binding quality, we also employed the widely adopted relative metric, the Min. < Re. ratio, to assess the quality of the generated binding conformations. This metric compares the pocket-binding quality of the generated conformation against the conformation obtained through classical force-field sampling. EDMolGPT demonstrated that 37% of its generated molecules exhibited binding modes superior to their force-field sampled counterparts, a performance that surpassed other models. Collectively, these results demonstrate that EDMolGPT generates ligands with notably better pocket-binding modes compared to baseline models.

**Results on Conformational Stability** As shown in Tab. 1, we evaluate the conformational stability through strain energy analysis. While ED2Mol employs Qscore-guided (Terwilliger et al., 2006) fragment placement, which penalizes deviations from ideal bond geometries and applies force-field refinement (smina (Ding et al., 2022a)) during generation, EDMolGPT does not employ any post-processing, yet still achieves comparable quality. Specifically, the strain energies of EDMolGPT-generated molecules are 33, 69, and 194 kcal/mol at the 25%, 50%, and 75% quantiles, which are on par with those of Pocket2Mol and Lingo3dMol. These results indicate that EDMolGPT encodes 3D structural constraints during generation, producing conformations with competitive stability.

**Results on Properties**   As shown in Tab. 1, we compare QED, SAS, and molecular weight across different methods. EDMolGPT achieves a strong balance across all three metrics, generating molecules with competitive QED and SAS values while maintaining molecular weights close to the Reference ligands—the ground-truth holo binders. This suggests that EDMolGPT produces chemically plausible molecules that reflect the size and complexity of real bioactive compounds. In contrast, some baselines, such as ED2Mol and ECloudGen, achieve favorable QED/SAS scores by generating smaller, simpler molecules or producing heavier but less chemically favorable structures. We also present a more detailed comparison between ED2Mol and EDMolGPT in Fig. 5. While ED2Mol attains higher QED and lower SAS, its molecules are generally much smaller, indicating that its apparent advantages stem from producing simpler structures rather than diverse, drug-like candidates, which limits its coverage of the chemical space relevant for realistic drug design.

Furthermore, we conduct a detailed comparison between our method, EDMolGPT, and ED2Mol by binning generated molecules according to molecular weight and evaluating drug-like properties within each bin. As shown in Tab. 2, the binned analysis shows that while Ed2Mol achieves higher QED in certain ranges, its SAS increases significantly with molecular weight, indicating reduced synthetic accessibility for larger molecules. In contrast, EdMolGPT maintains more stable and favorable SAS across all weight bins. Moreover, considering metrics such as recovery rate and binding affinity, EdMolGPT demonstrates stronger overall performance in generating biologically relevant structures. We acknowledge that Ed2Mol has strengths in optimizing specific drug-likeness properties, and view it as a valuable benchmark. In future work, we plan to incorporate these advantages to further improve the chemical quality of our generated molecules.

## 4.3. Results on ExpED

As shown in Tab. 3, we report the performance under the ExpED setting. Although the docking score (Min-in-place) appears relatively modest, this metric does not fully capture the binding potential in scenarios involving flexible protein conformations. Unlike pocket-based methods that assume a rigid binding site, EDMolGPT naturally accounts for conformational variability. Therefore, EDMolGPT under the ExpED setting can generate bioactive ligands that lie outside the reachable chemical space of generative models or virtual screening pipelines operating on rigid-pocket representations, as such ligands would be rejected a priori due to steric clashes. As shown in Fig. 6, the region highlighted in purple indicates a steric clash under a rigid pocket assumption, whereas experimental observations suggest this region is flexible and can accommodate the generated lig-

*Table 1.* The results with Binding Affinity, Conformation Stability (Conf. Stability), and Bioactive Molecule Recovery (Bio. Mol. Recov. ) on CalED, comparison across different methods. ↓ and ↑ indicate larger/smaller is better. Note: Min. < Re. denotes Min-in-place < Redocking. * *ED2Mol explicitly applies refinement utilizing external force fields before outputting molecules, a strategy not adopted by other models. This approach directly mitigates strain energy within the generated molecules.*

| Methods | Bio. Mol. Recov. | Binding Affinity | | | Conf. Stability | | | Drug-like Properties | | |
|---|---|---|---|---|---|---|---|---|---|---|
| | ECFP4_TS > 0.5 ↑ Ratio | Min-in-place ↓ Average | Redocking ↓ Average | Min. < Re. ↑ Ratio | Strain Energy ↓ 25% | 50% | 75% | QED ↑ Average | SAS ↓ Average | Weight Average |
| Pocket2Mol | 8% | -6.7 | -7.5 | 17.9% | 69 | 126 | 230 | 0.56 | 3.5 | 386 |
| TargetDiff | 3% | -6.2 | -7.0 | 15.2% | 139 | 313 | 643 | 0.60 | 4.0 | 299 |
| Lingo3DMol | 33% | -6.8 | **-7.8** | 12.0% | 13 | 40 | 177 | 0.59 | 3.1 | 348 |
| MolCRAFT | 17% | -6.1 | -6.9 | 20.1% | 20 | 47 | 102 | 0.51 | 3.81 | 285 |
| ED2Mol | 3% | -5.22 | -6.15 | 7.4% | 7* | 24* | 57* | **0.73** | 3.9 | 234 |
| ECloudGen | 6% | - | - | - | - | - | - | 0.66 | **2.9** | 213 |
| ECloudGen† | 33% | - | -6.68 | - | - | - | - | **0.73** | **2.9** | 326 |
| EDMolGPT | **41%** | **-6.92** | -7.18 | **37%** | 33 | 69 | 194 | 0.57 | 3.79 | 385 |
| Reference | **-** | -7.93 | -7.93 | - | - | - | - | 0.46 | 3.6 | 438 |

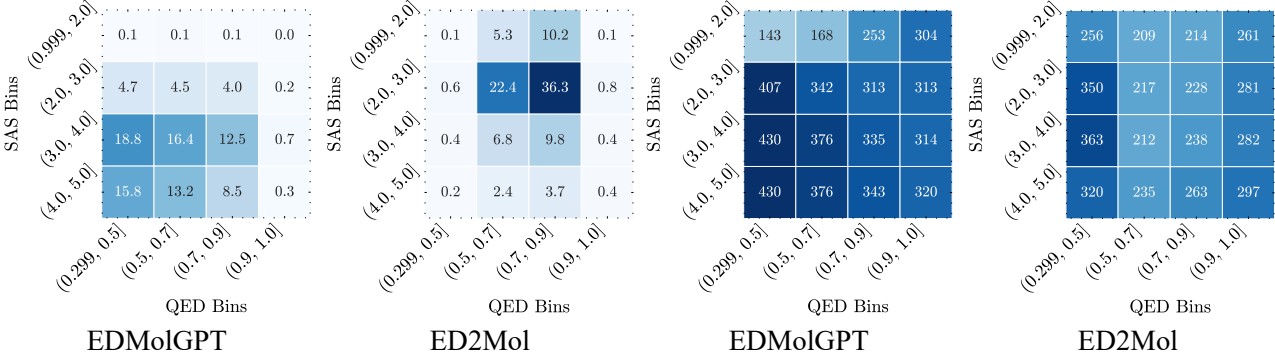

(a) Percentage of Samples by QED and SAS Bins      (b) Average Mol. Weight by QED and SAS Bins

*Figure 5.* The comparison between ED2Mol and EDMolGPT on QED, SAS, and Molecule Weight. We split QED and SAS into several bins and report the (a) Percentage of Samples by QED and SAS Bins and (b) Average Molecule Weight by QED and SAS Bins.

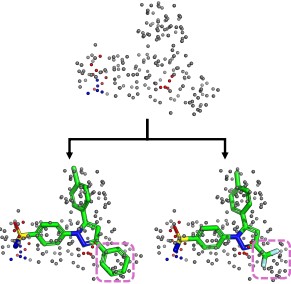

*Figure 6.* Generation from ED. Left: reproduction of the original ligand from ED. Right: a newly generated molecule that overlaps with the rigid pocket yet remains active.

and. The resulting molecule exhibits confirmed bioactivity, underscoring the limitations of rigid docking evaluations. More analyses are provided in Appendix Sec. B.

### 4.4. Ablation studies

**Ablations studies on resolution** $d_{min}$    To robustly enable our model to explore a broader chemical space and generate diverse scaffolds, differing from the reference ligand while still being guided by the binding environment, we utilize

*Table 2.* Comparison between ED2Mol and EDMolGPT across different molecular weight bins.

| Mol. Weight | SAS (ED2Mol / Ours) | QED (ED2Mol / Ours) |
|---|---|---|
| < 180 | 3.18 / 3.29 | 0.66 / 0.52 |
| 180–300 | 3.93 / 3.80 | 0.75 / 0.63 |
| 300–420 | 4.68 / 3.73 | 0.74 / 0.63 |
| > 420 | 5.27 / 3.88 | 0.52 / 0.46 |

*Table 3.* The results on ExpED.

| | Min-in-place | Recov. | QED | SAS | Mol. Weight |
|---|---|---|---|---|---|
| EDMolGPT | -5.4 | 20% | 0.50 | 3.69 | 372 |

low-resolution ED point clouds of reference binders. In our implementation, these low-resolution ED point clouds are achieved through two controls: the diffraction resolution ($d_{min}$), which determines the coarseness of the electron density, and the ED grid resolution, governed by the number of sampling points ($N_p$), which dictates the sparsity of the point cloud. Crucially, our ablation studies on diffraction resolutions (Tab. 4) indicate that $N_p$ plays a more significant role in constructing this low-resolution representation. Specifically, setting $N_p = 199$ consistently generates ED point cloud representations that mitigate bias associated

*Table 4.* Ablation studies on temperature and resolution. Div denotes the average score of ECFP_TS similarity between generated molecules and reference ligands, reflecting the structural diversity of the generated scaffolds.

| $d_{\min}$ | $T$ | Min-in-place | Redock | Min. $<$ Re. | Recov. | Div |
|---|---|---|---|---|---|---|
| 1.5Å | 0.7 | -6.94 | -7.12 | 33% | 46% | 0.186 |
| | 1.2 | -6.90 | -7.21 | 36% | 44% | 0.178 |
| 3.5Å | 0.7 | -6.92 | -7.18 | 37% | 41% | 0.184 |
| | 1.2 | -6.91 | -7.17 | 37% | 41% | 0.176 |

with the reference binders from which the low-resolution ED point clouds are derived, irrespective of the chosen diffraction resolution. This is supported by the consistently low Tanimoto similarity (typically $< 0.2$) between the generated molecules and the initial reference binders, as shown in Tab. 4 Div metric. Furthermore, varying the resolution does not introduce notable changes to the evaluation metrics.

**Ablation studies on temperature**   Another key hyperparameter in EDMolGPT is the sampling temperature $T$, which controls the randomness of the autoregressive generation process. Lower temperatures tend to produce more deterministic outputs, while higher temperatures encourage greater diversity but can introduce noisier conformations. As shown in Tab. 4, increasing $T$ from 0.7 to 1.2 has a more noticeable impact on the Div score, which decreases slightly, indicating a modest reduction in scaffold diversity. Other metrics change only marginally: redocking scores improve slightly and the fraction of cases where the redocking score is lower than the minimum-in-place score increases, suggesting better pocket alignment, while the recovery rate drops modestly. These results show that sampling temperature significantly influences the balance between diversity and structural consistency in generation.

## 5. Conclusion

In this work, we present EDMolGPT, a novel decoder-only autoregressive framework for 3D drug design that leverages electron-density–derived point clouds as conditioning signals. By sampling low-resolution electron density from an existing binder instead of relying on rigid pocket representations, our approach flexibly describes the binding environment, enabling the generation of ligands with chemically plausible conformations. Experiments on over 100 DUD-E targets show EDMolGPT outperforms existing structure-based generative methods in both 3D and 2D chemical spaces, with improved binding modes and higher recovery rates of bioactive molecules. EDMolGPT offers a paradigm to advance current technologies in drug design.

## Impact Statement

This paper presents work whose goal is to advance the field of Machine Learning. There are many potential societal consequences of our work, none which we feel must be specifically highlighted here.

## Acknowledgements

This work was supported in part by the National Natural Science Foundation of China No. 62376277, Public Computing Cloud, Renmin University of China, and fund for building world-class universities (disciplines) of Renmin University of China.

This work was also supported by the Beijing Municipal Science and Technology Commission (grant no. Z241100007724005 received by B.H.).

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

## Supplementary material

## Overview

This appendix presents comprehensive experimental details, evaluation details, and more visualization results. The content is organized into five main sections:

- Sec. A discusses the practical significance of electron density–based generation and contrasts it with pocket-based generation.

- Sec. B discusses the differences and meaning of applying ED for drug design.

- Sec. C.1, Sec. C.2, and Sec. C.3 provide further details on FSMILES, discretized 3D coordinates, and relative distances, supplementing the main text with specifications of the molecular input format.

- Sec. E.1, Sec. E.2 and Sec. E.3 provide additional details on the DUD-E dataset, the hyper-parameters of EDMolGPT, and more explanation for Bioactive Molecule Recovery metric.

- Sec. D explains how relative distance is employed to constrain the prediction of discretized 3D coordinates.

- Sec. F.1 examines the differences in data distributions between the public dataset used to train EDMolGPT and the DUD-E dataset, while Sec. F.2 presents additional visualization results. Sec. F.3 presents the computational efficieny between different methods. Sec. F.4 presents the ablation studies on $N_p$ and $c_p$. Sec. F.5 presents an analysis of binding flexibility beyond rigid docking assumptions.

## A. Discussion

Unlike previous structure-based generative approaches that explicitly rely on the precise 3D geometry of the protein pocket and often operate under a rigid pocket assumption, EDMolGPT leverages low-resolution electron density, derived from pre-existing binders within the pocket, to guide molecular generation.

The broad applicability of our method in practical drug discovery scenarios is evident from two key perspectives. Firstly, it is highly applicable whenever a protein structure has been solved with a binder in the pocket, a common occurrence in the early stages of drug discovery. Observing a binder within a potential binding site is frequently a prerequisite for confirming its designation as a 'pocket' and for establishing its holo-state for subsequent drug screening or design. This holds true regardless of whether the binder is a natural cofactor (e.g., NADPH or ADP) or a reference tool compound, all of which our model can utilize. Secondly, our method's versatility extends to scenarios where the binding pocket structure itself has not been experimentally solved. In such instances, if an active compound is known, its major conformations in aqueous solution can be determined through molecular dynamics simulations. These computationally derived conformations can then be utilized for the construction of low-resolution electron density, which subsequently fuels our method. This approach is supported by prior research indicating that highly active compounds tend to exhibit minimal differences between their dominant solution-phase conformations and their bound conformations, thereby incurring low strain energy upon binding (Tong & Zhao, 2021; Gu et al., 2021). While our method, to some extent, shares conceptual similarities with Ligand-Based Drug Design (LBDD), it offers broader applicability and distinct advantages. Current LBDD techniques typically require a series of ligands with a wide range of activities to establish Structure-Activity Relationships (SAR) and primarily focus on 2D molecular generation based on molecular fingerprints. In contrast, EDMolGPT offers a much simplified input requirement and provides unique strengths in the 3D generation space.

Considering that over 96% of clinical trials in practical drug discovery pipelines focus on previously studied targets with either known ligand or target information (Vasan et al., 2023), EDMolGPT is well-positioned to address the majority of real-world drug discovery cases, offering a complementary paradigm to conventional pocket-structure-based design methods.

## B. More discussions about ExpED and CalED

### B.1. Flexibility

Most current AI molecule generation models for SBDD overlook the dynamic nature of binding sites by assuming a static pocket representation (Feng et al., 2024; Qu et al., 2024). As shown in Fig. 7, modeling the pocket as a rigid

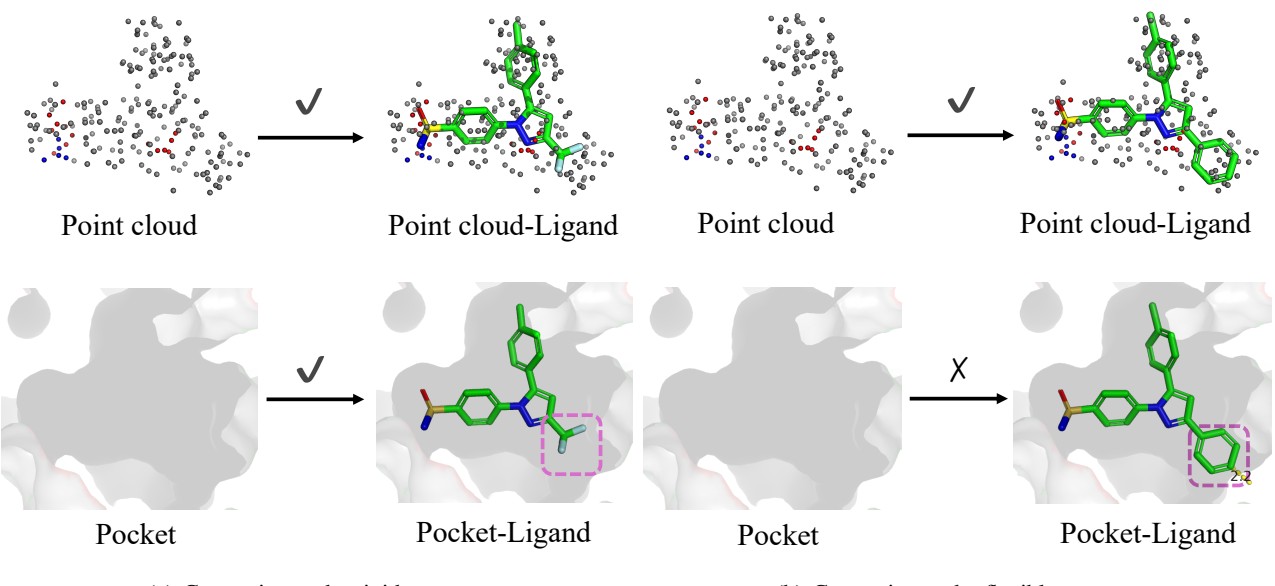

(a) Generation under rigid structure

(b) Generation under flexible structure

*Figure 7.* The generation pipeline leverages two distinct inputs: the static crystallographic pocket (PDB ID: 3L1N) and low-resolution electron density. (a) A ligand recovered by both structure- and density-guided approaches. (b) A ligand exclusively generated through electron density guidance. The region highlighted in purple illustrates an apparent steric clash between the generated phenyl moiety and the rigid conformation of the pocket.

structure fails to capture the intrinsic flexibility of proteins and their conformational changes upon ligand binding (Lu et al., 2024). Among various candidate representations, ED emerges as a highly informative descriptor, inherently encoding the binding site's spatial distribution, its physicochemical environment, critical intermolecular interactions, and crucial pocket flexibility. Leveraging electron density as a conditional input allows models to capture subtle steric, electrostatic, and non-covalent interaction features across multiple conformational states, enabling the generation of molecules that are not only geometrically compatible but also chemically and biologically meaningful (Ding et al., 2022b; Ma et al., 2023). As shown in Fig. 7, biochemical assays confirm the activity of ligands bearing bulky substituents at this site, indicating that the binding site is conformationally flexible and not comprehensively represented by the static structure. Accordingly, generation constrained by the static pocket fails to produce these active compounds. In contrast, our low-resolution ED–guided approach accommodates local conformational plasticity, enabling the successful generation of bulky, yet active, substituents.

### B.2. Difference between Exped and CalED

As shown in Fig. 8, the intensity distributions of ExpED and CalED differ noticeably. ExpED generally exhibits lower and more variable densities compared with CalED, reflecting experimental noise and structural flexibility. These distributional discrepancies motivate our two-stage training strategy: the model is first pre-trained on large-scale CalED, benefiting from stable and abundant computed densities, and then fine-tuned on ExpED to adapt to the characteristics of experimental measurements.

## C. Representation of Molecule

### C.1. FSMILES

The original FSMILES was developed for the autoregressive task in Lingo3DMol. The pre-defined token vocabulary is shown as follows:

```
fsmiles_list = [
            "pad_0", "start_0", "end_0", "sep_0",
            "C_0", "C_5", "C_6", "C_10", "C_11", "C_12",
```

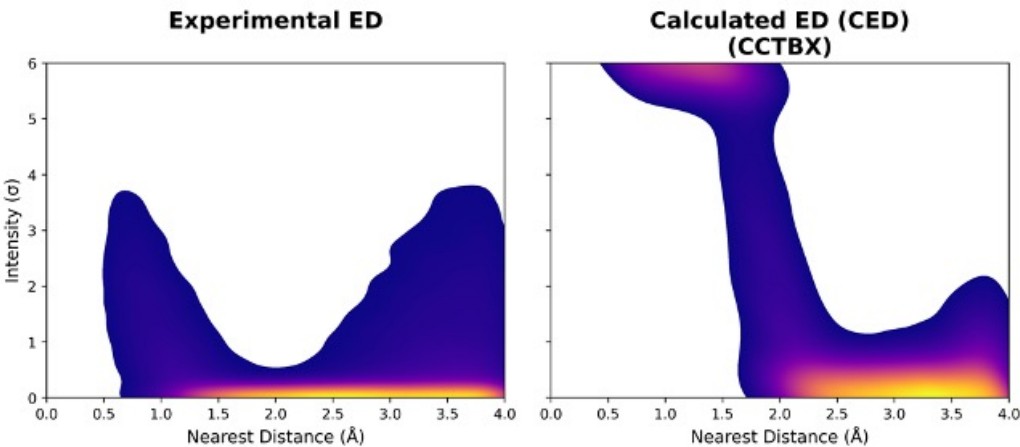

*Figure 8.* Comparison of ED intensity distributions across different calculation methods. From left to right: (a) ExpED (b) CalED

```
        "c_0", "c_5", "c_6", "c_10", "c_11", "c_12",
        "N_0", "N_5", "N_6", "N_10", "N_11", "N_12",
        "n_0", "n_5", "n_6", "n_10", "n_11", "n_12",
        "S_0",
        "s_0", "s_5", "s_6", "s_10", "s_11", "s_12",
        "O_0", "O_5", "O_6", "O_10", "O_11", "O_12",
        "o_0", "o_5", "o_6", "+_0", "o_11", "o_12",
        "F_0",
        "Cl_0",
        "[nH]_0", "[nH]_5", "[nH]_6",
        "[nH]_10", "[nH]_11", "[nH]_12",
        "Br_0",
        "/_0", "\\_0", "@_0", "@@_0", "H_0",
        "1_0", "2_0", "3_0", "4_0", "5_0", "6_0",
        "#_0", "=_0", "-_0", "(_0", ")_0",
        "[_0", "]_0", "[*]_0","([*])_0"
    ].
```

In FSMILES, the tokens pad_0, start_0, end_0, and sep_0 denote the padding token, the start-of-molecule marker, the end-of-molecule marker, and the fragment separator, respectively. However, the current FSMILES implementation, which is based on BRICS (Degen et al., 2008), often produces an excessive number of small, highly fragmented substructures. To address this limitation, we refine the FSMILES decomposition process by introducing a fragment-consolidation strategy. Specifically, we preserve any bond whose cleavage would generate fragments containing fewer than three atoms, thereby mitigating over-fragmentation. As illustrated in Fig. 9, our method prevents unnecessary segmentation. For example, in the left panel, the hydroxyl group (-OH) is cleaved in the original FSMILES decomposition, whereas our approach preserves it, resulting in a more chemically meaningful fragment.

For the better understanding of FSMILS, we also visualize the density of fragment atom count and the density of fragment molecular weight. As shown in Fig. 10, the majority of fragments contain a moderate number of atoms, and the molecular weights are concentrated within a reasonable range, indicating that the fragment decomposition produces chemically meaningful substructures. These distributions suggest that our fragmentation strategy successfully balances the granularity of molecular breakdown with the preservation of structurally relevant motifs, avoiding excessive over-fragmentation while maintaining fragments suitable for downstream modeling.

## C.2. Discretized 3D coordinates

We provide further clarifications regarding the discretization scheme described in the main text.

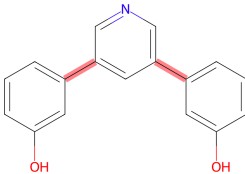

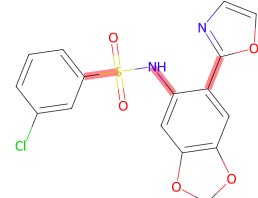

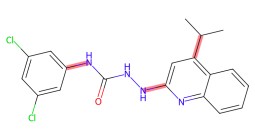

start_0O_0c_61_0c_6c_6c_6c_6([*])
_0c_61_0sep_0c_61_0c_6n_6c_6c_
6([*])_0c_61_0sep_0c_61_0c_6c_6c
_6c_6(_0O_0)_0c_61_0sep_0end_0

start_0O_0=_0S_0([*])_0(_0=_0O_0)
_0N_0[*]_0sep_0c_61_0c_6c_112_0
c_11(_0c_6c_61_0[*]_0)_0O_5C_5O
_52_0sep_0c_51_0n_5c_5c_5o_51_
0sep_0c_61_0c_6c_6c_6c_6(_0Cl_0
)_0c_61_0sep_0end_0

start_0C_0C_0([*])_0C_0sep_0c_61
_0c_6c_6([*])_0n_6c_122_0c_6c_6c
_6c_6c_121_02_0sep_0N_0N_0C_0
(_0=_0O_0)_0N_0[*]_0sep_0c_61_0
c_6c_6(_0Cl_0)_0c_6c_6(_0Cl_0)_0
c_61_0sep_0end_0

(a) FSMILES

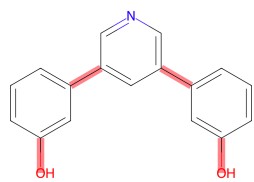

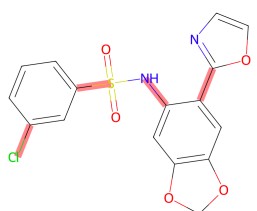

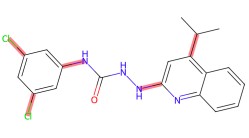

start_0O_0[*]_0sep_0c_61_0c_6c_6
c_6c_6([*])_0c_61_0sep_0c_61_0c_
6n_6c_6c_6([*])_0c_61_0sep_0c_61
_0c_6c_6c_6([*])_0c_61_0sep_0
O_0sep_0end_0

start_0O_0=_0S_0([*])_0(_0=_0O_0)
_0N_0[*]_0sep_0c_61_0c_6c_112_0
c_11(_0c_6c_61_0[*]_0)_0O_5C_5O
_52_0sep_0c_51_0n_5c_5c_5o_51_
0sep_0c_61_0c_6c_6c_6c_6([*])_0c
_61_0sep_0Cl_0sep_0end_0

start_0C_0C_0([*])_0C_0sep_0c_61
_0c_6c_6([*])_0n_6c_122_0c_6c_6c
_6c_6c_121_02_0sep_0N_0N_0C_0
(_0=_0O_0)_0N_0[*]_0sep_0c_61_0
c_6c_6([*])_0c_6c_6([*])_0c_61_0se
p_0Cl_0sep_0Cl_0sep_0end_0

(b) Ours

*Figure 9.* The comparison between (a) FSMILES and (b) Ours. We highlight the cut bonds in red, and the tokenized result is marked below.

First, the choice of resolution $\sigma = 0.1$ Å reflects a trade-off between geometric fidelity and vocabulary size. With this setting, the maximum quantization error per coordinate dimension is $0.05$ Å, which is negligible compared to the typical bond length in organic molecules ($\sim 1.2$–$1.5$ Å). Thus, the discretized representation is sufficiently accurate to preserve chemically meaningful structures.

Second, the positive shift applied after discretization ensures that all coordinates $\widehat{v}_m^i$ lie within the range of non-negative integers. This design is important because it allows us to directly map each integer triplet to a unique token in the model vocabulary without introducing negative indices, thereby simplifying both tokenization and embedding lookups.

Third, although discretization maps continuous space into a bounded integer lattice, the autoregressive model does not rely solely on absolute coordinate tokens. Instead, the generation process is conditioned on relative geometric features (bond length $l$, bond angle $\theta$, and dihedral angle $\phi$), which act as local structural constraints during inference. This hybrid formulation mitigates the potential artifacts of quantization by guiding the coordinate recovery step toward chemically valid regions.

Finally, both molecular coordinates and auxiliary point clouds are discretized into the same shifted lattice space. This alignment enables us to encode heterogeneous geometric information (e.g., atomic positions and electron-density-derived point clouds) within a unified tokenization framework, which greatly facilitates multimodal training.

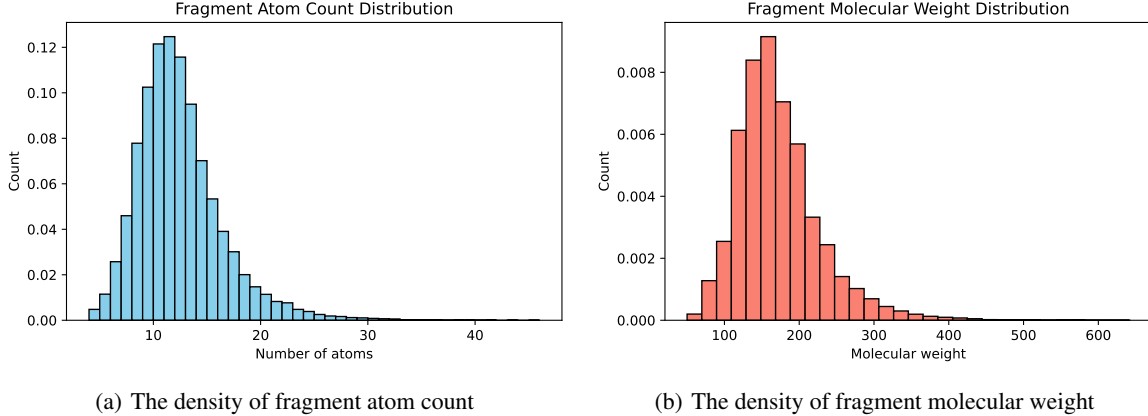

(a) The density of fragment atom count  (b) The density of fragment molecular weight

*Figure 10.* The visualization results on (a) Fragment Atom Count Distribution and (b) Fragment Molecular Weight Distribution.

## C.3. Relative distance

To determine the reference atoms required for autoregressive coordinate generation, we design a procedure to trace the ancestral nodes of each token in the molecular sequence $\mathcal{M} = \{a_m^1, a_m^2, \ldots, a_m^n\}$. For a given step $i$, we define three levels of ancestor indices:

$$r_1(i), \quad r_2(i) = r_1(r_1(i)), \quad r_3(i) = r_1(r_2(i)), \tag{9}$$

where $r_1(i)$ denotes the *first-order ancestor*, $r_2(i)$ the *second-order ancestor*, and $r_3(i)$ the *third-order ancestor* of token $a_m^i$.

The search for $r_1(i)$ is performed by traversing the sequence backward from step $i$: (1) if $a_m^i$ is an atom token, $r_1(i)$ is assigned to the nearest preceding atom token; (2) if $a_m^i$ is a non-element symbol (e.g., branch markers "(" and ")"), we recursively skip bracketed fragments while ensuring valid pairing of parentheses, thus locating the chemically valid attachment point of the current substructure; (3) if a separator token sep_0 is encountered, the search crosses fragment boundaries and jumps to the most recent star marker $[\cdot]$ that denotes a fragment connection point.

Once $r_1(i)$ is determined, higher-order ancestors are obtained recursively as

$$r_2(i) = r_1(r_1(i)), \qquad r_3(i) = r_1(r_2(i)). \tag{10}$$

These indices provide the hierarchical reference atoms for step $i$, which correspond to the spatial coordinates

$$\boldsymbol{v}_m^{i-1} = \boldsymbol{v}_{r_1(i)}, \quad \boldsymbol{v}_m^{i-2} = \boldsymbol{v}_{r_2(i)}, \quad \boldsymbol{v}_m^{i-3} = \boldsymbol{v}_{r_3(i)}. \tag{11}$$

In this way, the algorithm ensures that each new atom position $\boldsymbol{v}_m^i$ is generated with respect to three previously defined reference atoms, enabling consistent computation of bond length, bond angle, and dihedral angle during molecular construction.

## D. Details about inference

During inference, we feed the conditioned point cloud into EDMolGPT and generate the molecular sequence $\mathcal{M}$ in an autoregressive, token-by-token manner. For FSMILES tokens $\widehat{a}_m^i$ and relative geometric tokens $\widehat{l}_m^i, \widehat{\theta}_m^i, \widehat{\phi}_m^i$, we apply temperature sampling (Radford et al., 2019) to draw predictions from the model's output distribution. However, directly sampling discretized 3D coordinates $\widehat{\boldsymbol{v}}_m^i$ from the entire spatial domain often results in unrealistic or geometrically distorted molecular structures. To address this issue, we exploit the predicted relative geometric features to restrict the sampling space for $\boldsymbol{v}_m^i$.

Specifically, given the three previously generated atom positions $\boldsymbol{v}_m^{i-1}, \boldsymbol{v}_m^{i-2}, \boldsymbol{v}_m^{i-3}$, and the predicted discretized features $(\widehat{l}_m^i, \widehat{\theta}_m^i, \widehat{\phi}_m^i)$, we first recover the corresponding continuous values according to the discretization rule (cf. Eq. 6):

$$l_m^i = \widehat{l}_m^i \cdot \sigma, \quad \theta_m^i = \widehat{\theta}_m^i \cdot 10^\circ, \quad \phi_m^i = \widehat{\phi}_m^i \cdot 10^\circ. \tag{12}$$

*Table 5.* Hyperparameter settings for our EDMolGPT based model.

| Hyperparameter | Value |
|---|---|
| input_vocab_size | 300 |
| input_dist_size | 300 |
| num_bond_ang | 200 |
| num_bond_leng | 200 |
| num_dih_ang | 200 |
| n_layer | 24 |
| n_embd | 1024 |
| n_ctx (= n_positions) | 1024 |
| n_head | 16 |
| activation_function | gelu_new |
| resid_pdrop | 0.1 |
| embd_pdrop | 0.1 |
| attn_pdrop | 0.1 |
| layer_norm_epsilon | 1e-5 |
| initializer_range | 0.02 |

Instead of enforcing these values deterministically, we introduce tolerance intervals:

$$l_m^i \in [l_m^i - \delta_l, l_m^i + \delta_l,], \theta_m^i \in [\theta_m^i - \delta_\theta, \theta_m^i + \delta_\theta], \ \phi_m^i \in [\phi_m^i - \delta_\phi, \phi_m^i + \delta_\phi], \tag{13}$$

where $\delta_l$, $\delta_\theta$, and $\delta_\phi$ control the flexibility of bond length, bond angle, and dihedral angle, respectively (empirically, $\delta_l \approx 0.1$Å, $\delta_\theta \approx 10°$, $\delta_\phi \approx 10°$).

Accordingly, the feasible region of the next atom coordinate $v_m^i$ is constrained as

$$
\begin{aligned}
v_m^i \in \Big\{ v \in \mathbb{R}^3 \Big| & |v - v_m^{i-1}| \in [l_m^i - \delta_l, l_m^i + \delta_l], \\
& \theta(v) \in [\theta_m^i - \delta_\theta, \theta_m^i + \delta_\theta], \\
& \phi(v) \in [\phi_m^i - \delta_\phi, \phi_m^i + \delta_\phi] \Big\}.
\end{aligned}
\tag{14}
$$

This procedure constrains $v_m^i$ to a narrow spherical patch determined by the predicted local structure, thereby ensuring geometric consistency with the previously generated atoms while allowing moderate flexibility to account for discretization errors and model uncertainty.

## E. Experimental setting

### E.1. DUD-E dataset

The DUD-E (Directory of Useful Decoys: Enhanced) dataset is a large-scale benchmark designed for the development and evaluation of virtual screening algorithms. It is an enhanced version of the original DUD dataset, constructed by the Shoichet Laboratory at the University of California, San Francisco (UCSF). DUD-E comprises 102 protein targets, spanning diverse families such as kinases, proteases, GPCRs, nuclear receptors, and ion channels, thereby covering a broad spectrum of pharmacologically relevant classes. For each target, the dataset provides a curated set of experimentally validated active ligands together with approximately 50 property-matched decoys per active compound. These decoys are selected to mimic the actives in terms of simple physicochemical descriptors (e.g., molecular weight, hydrogen bond donors/acceptors, logP), but are topologically distinct, making them unlikely to bind the target. This careful design allows DUD-E to reduce dataset bias and to more reliably evaluate the discriminative power of computational screening methods. Consequently, DUD-E has become a standard benchmark for assessing the performance of molecular docking, machine learning–based virtual screening, and structure-based drug design approaches.

### E.2. Hyper-parameters of EDMolGPT

As shown in Tab. 5, we summarize the hyperparameter settings of our EDMolGPT model. The vocabulary size for FSMILES tokens and coordinate tokens are specified by input_vocab_size and input_dist_size, both set to 300, which we found

sufficiently large to accommodate current requirements while leaving room for future extensions. Similarly, the numbers of tokens representing bond lengths, bond angles, and dihedral angles (num_bond_leng, num_bond_ang, and num_dih_ang) are each set to 200, ensuring adequate coverage of structural variations with flexibility for expansion.

For the general architecture, we extend the GPT-2 backbone with n_layer = 24 transformer blocks, hidden dimension n_embd = 1024, context length n_ctx = 1024, and n_head = 16 attention heads, which provide higher model capacity suitable for molecular generation. The activation function is set to gelu_new (Hendrycks & Gimpel, 2016), while dropout is consistently applied at multiple levels (resid_pdrop, embd_pdrop, attn_pdrop, all 0.1) to mitigate overfitting. Other parameters, including the layer normalization epsilon (1e-5) and weight initialization range (0.02), follow the standard GPT-2 configuration to ensure stable optimization.

Overall, our design largely inherits the strengths of GPT-2 while incorporating domain-specific vocabulary extensions and scaling adjustments to better capture molecular structural information.

### E.3. Metrics for Bioactive Molecule Recovery

We use several metrics to evaluate the biological relevance and structural quality of our generated molecules. Among these, the ECFP4 Tanimoto Similarity (TS) is employed to specifically measure the ability of the model to generate molecules that exhibit high putative biological activity, which we detail here. ECFP4 (Extended Connectivity Fingerprints, diameter 4) is a widely accepted circular fingerprint in chemoinformatics that encodes the local chemical environment around each atom, capturing essential pharmacophoric features strongly correlated with bioactivity. We evaluate this metric by comparing our generated molecules against a set of already-validated active molecules corresponding to the target pockets, which are sourced from the established DUDE (Database of Useful Decoys: Enhanced) benchmark. A high ECFP4 TS score between a generated molecule and a known active molecule indicates that the generated compound successfully recapitulates or is structurally analogous to a proven active scaffold. This measure serves as a crucial chemoinformatics-based validation and effectively complements purely physical metrics like docking scores, allowing us to holistically assess that the generated output is not only stable in the pocket but also chemically plausible as a bioactive compound.

## F. More experimental results

### F.1. Distribution analysis

To ensure that our method truly generates novel molecules rather than memorizing those from the training set, we conducted a distributional overlap analysis between the training and test data. The training set consists of approximately two million molecules curated from public databases with additional drug-likeness filtering. In contrast, the DUD-E dataset, used for evaluation, was independently processed into point cloud representations of active ligands. Since it is unclear whether any structural overlap exists between the DUD-E molecules and the training set, it is necessary to explicitly verify the degree of intersection. If a significant overlap were present, one could not rule out the possibility that our model merely recalls known molecules instead of generating meaningful new candidates.

To address this, we performed a nearest-neighbor retrieval experiment: for each active ligand point cloud from DUD-E, we searched the training set for the most similar point cloud according to the DICE coefficient, and recorded the corresponding similarity scores. Visualization of the results (Fig. 11) shows that the maximum DICE similarity never exceeds 60%. This low overlap confirms that the test set structures are not contained in the training data, thereby validating that our model's outputs represent genuine generation rather than memorization.

### F.2. Visualization of generation results

To further evaluate the effectiveness of our framework, we conducted visualization analyses on three representative protein–ligand complexes (1sj0, 3lan, and 2etr), as shown in Fig. 12. For each target, the electron density–derived point cloud was used as the conditioning input, and we compared the generated ligands against the experimentally determined ground truth. The visualizations demonstrate that our method produces ligands that align well with the spatial distribution of the point cloud, indicating that the generative process effectively captures the underlying geometric constraints of the binding pocket. In addition, the generated ligands consistently achieve favorable docking scores, often lower than those of the reference structures, suggesting strong binding compatibility and chemical plausibility. Importantly, across all three systems, our framework is able to generate multiple diverse ligand candidates while maintaining close agreement with

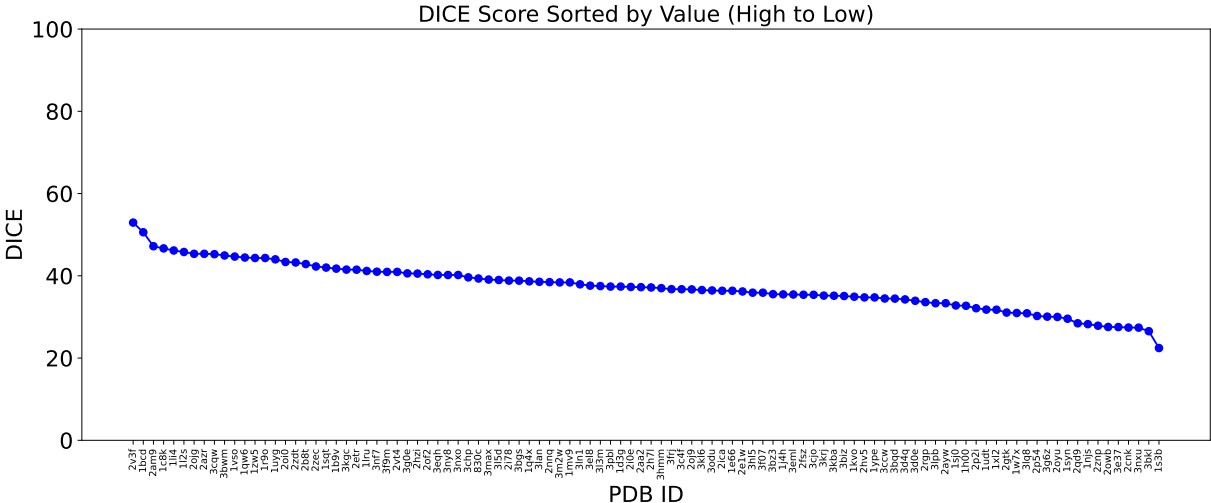

*Figure 11.* DICE similarity scores between DUD-E active ligands and their closest counterparts in the training dataset, sorted from high to low. Each point corresponds to a DUD-E ligand, with the horizontal axis indicating the PDB ID and the vertical axis showing the maximum DICE score identified in the training set. The results indicate that all maximum DICE scores remain below 60%.

the pocket environment. This combination of low docking scores and structural diversity highlights the rationality of our design and suggests that the method can reliably explore alternative binding modes without sacrificing physical or chemical feasibility.

### F.3. Computational efficiency

As shown in Tab. 6, EDMolGPT utilizes an autoregressive GPT architecture conditioned by the electron density map, and achieves a competitive average generation speed of approximately 1.5 seconds per molecule. For comparison, we have compiled the reported or estimated generation speeds for several prominent SBDD models. Pocket2Mol, another autoregressive model, reports a highly optimized generation speed of approximately 0.45 seconds per molecule. For diffusion-based models like TargetDiff, the speed depends heavily on the number of sampling steps (TargetDiff uses 1000 steps); the multi-step nature of the diffusion process typically makes them significantly slower than highly optimized autoregressive models.

*Table 6.* Comparison of molecular generation speed across different models.

| Model | Architecture | Avg. Generation Time (s/molecule) |
|---|---|---|
| ED-GPT | Autoregressive | $\approx 1.5$ |
| Pocket2Mol | Autoregressive | $\approx 0.45$ |
| TargetDiff | Diffusion | $\approx 7$ |

### F.4. Ablation studies on $N_p$ and pharmacophore labels $c_p$

As shown in Tab. 7, we perform ablation studies on $N_p$ and the use of pharmacophore labels $c_p$, which are key hyperparameters for modeling the electron density. Increasing $N_p$ provides a more detailed description of the positive electron-density

*Table 7.* Ablation results on $N_p$ and pharmacophore labels.

| | Min-in-place | Div |
|---|---|---|
| $N_p = 100$ | -6.46 | 0.15 |
| $N_p = 300$ | -7.22 | 0.20 |
| w/o $c_p$ | -6.15 | 0.09 |

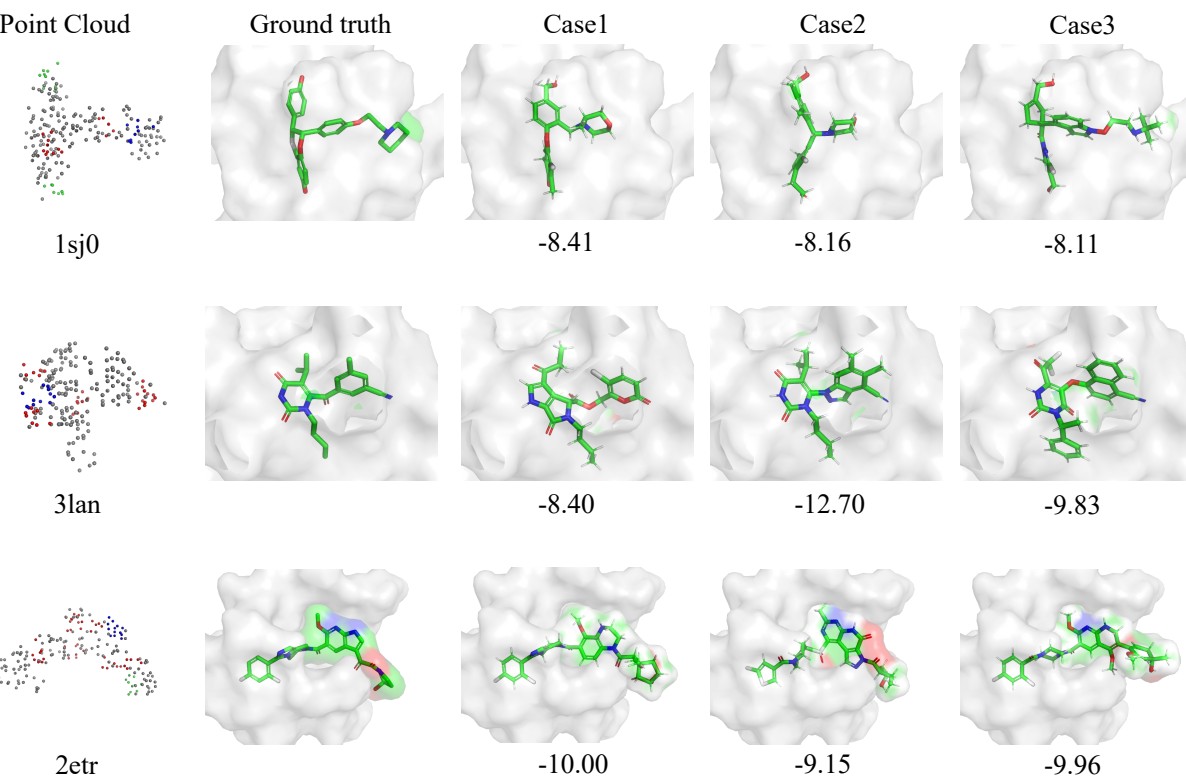

*Figure 12.* Visualization of three protein–ligand complexes with PDB IDs 1sj0, 3lan, and 2etr. The first column shows the point cloud extracted from the electron density map. The second column presents the ground-truth ligand conformations within the corresponding protein pockets. The following three columns (Case 1–3) display ligands generated by our method, with the associated minimum in-place docking scores indicated below each case.

*Table 8.* Number of targets for which recovered bioactive molecules obtain unfavorable Glide scores ($> -5$).

| Method | Targets with Glide Score $> -5$ |
| --- | --- |
| ED2Mol | 0 |
| EDMolGPT | 2 |

patterns, improving the Min-in-place score while slightly reducing diversity. Conversely, removing pharmacophore labels $c_p$ relaxes geometric constraints, increasing diversity but lowering Min-in-place scores. These results illustrate the expected trade-offs and confirm that both $N_p$ and pharmacophore guidance play important roles in ensuring the quality and structural fidelity of the generated molecules.

### F.5. More evaluation metrics for rigid pocket assumption

To further evaluate the ability to capture binding flexibility beyond rigid docking assumptions, we analyze Bioactive Molecule Recovery rather than relying solely on GlideSP scores. Specifically, for both EDMolGPT and ED2Mol, we identify generated molecules with Tanimoto similarity greater than 0.5 to known bioactive compounds for each target. We then perform Glide docking and count the number of targets whose recovered bioactive molecules obtain Glide scores higher than $-5$, which are generally considered unfavorable under rigid-pocket assumptions.

As shown in Tab. 8, EDMolGPT recovers bioactive molecules for two targets that still receive poor Glide scores, whereas ED2Mol recovers none. Since these molecules are experimentally known actives despite unfavorable rigid docking evaluations, the results suggest that EDMolGPT is capable of exploring binding patterns and conformational flexibility that are not adequately captured by rigid docking protocols. This provides additional evidence that our method can partially overcome the limitations imposed by rigid-pocket assumptions.

