# OpenReview forum: "From Holo Pockets to Electron Density: GPT-style Drug Design with Density"
_ICML.cc/2026/Conference — ICML 2026 regular_

### Official Review · Reviewer_hbu3 · 2026-03-10

**Soundness:** 2
**Presentation:** 4
**Significance:** 3
**Originality:** 3
**Overall Recommendation:** 4
**Confidence:** 3

**Summary:**

This paper introduces EDMolGPT, a decoder-only autoregressive model for structure-based drug design that conditions on electron density from the filler rather than the empty protein pocket. The motivation is straightforward: in real drug discovery you usually start with a holo complex where a ligand is already bound, and the filler's electron density captures conformational flexibility better than a rigid empty pocket would. The model uses two types of ED input: calculated ED (CalED) from atomic coordinates via FFT for pre-training on ~2M structures, and experimental cryo-EM/X-ray ED for fine-tuning on PDBbind. Molecules are generated token-by-token using a modified FSMILES representation that includes 3D geometric features like bond lengths and angles, with beam search constrained to physically reasonable geometries. On DUD-E, the method achieves competitive result compared to existing baselines.

**Compliance With Llm Reviewing Policy:**

Affirmed.

**Final Justification:**

After reading the rebuttal, my overall view is still moderately positive. The paper is clear, and the generation results are promising. However, I still have some concern about possible reliance on the conditioned filler state, and I also think the evaluation is somewhat limited in breadth. Overall, I think this is a good paper, but I still find the central motivation less compelling than I would like.

**Key Questions For Authors:**

1. A broader concern I have is whether this method is, to some extent, performing template-driven generation from the conditioned filler state rather than learning more generalizable binding principles. In Table 3, the ablation on point cloud size shows that Np=10000 gives 41% recovery while Np=5000 gives 38%, while the diversity scores remain nearly unchanged (0.186 vs 0.184). Does this suggest that the additional points mainly help preserve or exploit the reference filler state, thereby improving recovery of known actives rather than enabling broader exploration of new chemical space?

2. Figure 5 shows some generated molecules for KITH. I'm curious about the failure cases: when the model generates molecules with low binding scores or poor recovery, is there a pattern? For instance, does it happen more often when the reference ligand has unusual features, or when the electron density is noisy?

3. The two-stage training uses CalED for pre-training and ExpED for fine-tuning. Have you tried the reverse: pre-training on ExpED (even with less data) and then adapting to CalED? This might reveal whether the performance difference between CalED and ExpED comes from the density quality itself or from the amount of training data.

4. The comparison with ECloudGen shows your method performs better. But ECloudGen also uses electron density as input. What do you think is the key difference that leads to the improvement?

**Limitations:**

yes

**Strengths And Weaknesses:**

**Strengths**

1. The core idea of using filler electron density instead of empty pocket geometry is interesting.  Conditioning on the filler's ED captures conformational information that gets lost when you remove the ligand.

2. The 41% bioactive recovery rate is strong compared to the baselines. The generated molecules also show reasonable binding poses.

3. The paper is clearly written and the figures help a lot, the overall presentation makes the method easy to follow.

4. Overall, the paper stays focused on its main theme and the experiments are conducted around that theme. The motivation has value.

**Weaknesses**

1. Since the filler ED includes the ligand itself, the model is conditioning on information about the reference binder's shape and position. This raises a question about how much the model learns general binding principles versus relying on the reference ligand's geometry. A more detailed analysis of this aspect would be helpful for understanding the method better and would benefit the interpretation of the results. Additionally, I wonder if this could be viewed as a form of information leakage, or perhaps a trade-off where diversity is sacrificed for improved stability. I'm open to discussing this with the authors to better understand their perspective.

2. The ExpED results show a significant drop compared to CalED. The paper attributes this to conformational flexibility making rigid docking less reliable, which might be true, but there's no baseline comparison in Table 2 to contextualize whether this is expected or a limitation of the method.

3. For the binding affinity evaluation, why not include standard Vina docking scores? Table 1 shows min-in-place and recovery metrics, but a direct comparison using Vina would make it easier to assess performance against other SBDD methods that report those numbers.

4. All evaluation is conducted on DUD-E. While this is a standard benchmark, the reliance on a single dataset makes it harder to assess how broadly the method generalizes. Moreover, since DUD-E was originally designed as a decoy dataset for virtual screening, I'm somewhat concerned about whether it's the most appropriate choice for evaluating de novo generation methods.

---

> ### Author Rebuttal · Authors · 2026-03-31
>
> We appreciate your feedback. We promise that we will add all contents to the original paper.
> >weakness1
>
> ***response:*** As shown in Tab. 3, generated molecules have low structural similarity to reference ligands, demonstrating exploration beyond known binders. Appendix F.1 further quantifies spatial similarity between training and test densities using the DICE metric. During ExpED inference, the model incorporates solvent-derived electron density, excluded during training, forcing generalization to unseen density distributions rather than memorization. Moreover, ablations in Appendix F.4, varying sampling points and hydrogen-bond labels, show performance changes, confirming the model can capture fine-grained, physically meaningful features with increased representation capacity.
> >weakness2
>
> ***response:*** This decline reflects expected limitations: the scarcity of experimentally determined Holo structures restricts training data, and the higher flexibility and noise in experimental density complicate evaluation. While CalED pre-training helps, it cannot fully offset these challenges. As a pioneering effort, these results highlight the hurdles of moving from synthetic to real data. Future work will refine the model and adopt more robust scoring functions.
> >weakness3
>
> ***response:*** We use Glide for binding affinity evaluation as it is an industry-standard tool with a sophisticated scoring scheme that models hydrophobic enclosure, hydrogen bonding, and desolvation more accurately than simpler functions like Vina. Glide is widely adopted in prior literature [a, b] and provides better correlation with experimental affinities, making it suitable for assessing fine-grained molecular quality. To ensure reproducibility, we will release the processed electron density data and evaluation scripts upon publication.
>
> [a] Comparative assessment of scoring functions: the CASF-2016 update. J. Chem. Inf. Model.
>
> [b] Beware of the generic machine learning-based scoring functions in structure-based virtual screening. Brief. Bioinform.
> >weakness4
>
> ***response:*** We clarify that DUD-E is a suitable benchmark for generative evaluation. It covers over 100 diverse protein targets, including Kinases, Proteases, GPCRs, and Ion Channels, and all target structures and active compounds are experimentally determined. DUD-E consists of targets, experimentally measured actives, and synthetic decoys; for our evaluation, we only use targets and actives, ensuring that generated molecules are assessed against verified experimental data. These features make DUD-E particularly advantageous for evaluating de novo generation, and several recent generative works [c, d] also adopt it for benchmarking.
>
> [c] Generation of 3D molecules in pockets via a language model. Nature Machine Intelligence
>
> [d] Population-based de novo molecule generation, using grammatical evolution. Chemistry Letters
> >Question1
>
> ***response:*** Tab. 3 omits $N_p$ ablations, which are reported in Appendix F.4 (affinity and Div). We report the Bio. Mol. Recov. and Div for different $N_p$ values as follows:
> ||Bio. Mol. Recov.| Div|
> |---|---|---|
> |$N_p$=199|41%|0.18|
> |$N_p$=300|44%|0.2|
>
> Increasing $N_p$ improves the recovery rate while diversity remains nearly unchanged. It shows that the model explores more effective chemical space and generates molecules closer to known actives, without sacrificing diversity, verifying that our method captures meaningful chemical structure rather than overfitting.
> >Question2
>
> ***response:*** To analyze failure patterns, we define a failure as a positive docking score or a Glide preparation failure. Statistical evaluation shows that targets like 1s3b, 2znp, and 3ln1 have lower success rates, potentially correlated with molecular size: 1s3b’s reference ligands have high molecular weights (~969). Our fixed 199-point representation may struggle to capture larger ligands, suggesting representation density relative to binding site complexity contributes to failures.
>
> Failures are multifactorial, also involving point cloud shape and density quality; future work will investigate these factors to improve EDMolGPT’s robustness.
> >Question3
>
> ***response:*** The comparative results are summarized below:
> ||Bio. Mol. Recov.|Min-in-place|
> |---|---|---|
> |CalED|**41%**|**-6.92**|
> |ExpED$\to$ CalED|38%|-6.45|
>
> Recovery Rate is similar across both strategies, reflecting the robustness of the sequential model. However, affinity is lower for the reverse strategy, likely due to limited training on the large CalED dataset. The performance gap arises from a distribution shift between experimental and calculated densities (Appendix B.2) and the greater conformational flexibility of ExpED compared to rigid CalED.
> >Question4
>
> ***response:*** We acknowledge ECloudGen as inspiring, but unlike its sequence-based approach, our method directly models molecular structures, better capturing spatial constraints from electron density and improving structural fidelity.

---

> > ### Author Rebuttal · Reviewer_hbu3 · 2026-04-02
> >
> > Thank you to the authors for the detailed rebuttal and for engaging seriously with the questions. I appreciate the additional clarifications, and some of the responses are helpful. However, my main concern is only partially addressed, especially regarding the extent to which the model may rely on template-like information contained in the conditioned filler electron density, rather than learning more general principles. Some evaluation-related concerns are also only partially resolved.
> >
> > As a result, while I appreciate the authors’ effort, the rebuttal does not change my assessment enough to justify raising my score further. In my view, a score of 4 already reflects a positive evaluation for this paper and is not a low score.

---

> > > ### Author Response · Authors · 2026-04-02
> > >
> > > Thank you for your positive score and for your careful evaluation of our work.
> > >
> > > Regarding your suggestion to explore more general principles, we fully agree that this direction reflects the broader vision of AI-empowered drug design and is well aligned with our long-term goals. In this study, our method, leveraging experimental electron density to generate novel drug-like molecules, represents a step in this direction and provides a solid foundation for the eventual development of first-principles-based design strategies.  In future work, we plan to further advance this line of research by incorporating electron-density-topology-based molecular binding patterns, as well as electrostatic surface features, to better capture the underlying physical principles.
> > >
> > > If you have any additional suggestions, we would be very happy to discuss them in more detail. We truly value these exchanges and will respond as thoughtfully and thoroughly as possible.
> > >
> > > Thank you again for your support and encouraging feedback.

---

### Official Review · Reviewer_K9e2 · 2026-03-10

**Soundness:** 2
**Presentation:** 2
**Significance:** 3
**Originality:** 3
**Overall Recommendation:** 4
**Confidence:** 3

**Summary:**

The authors analyze a pertinent challenge in structure-based drug design (SBDD): existing methods typically condition molecule generation on empty binding pockets from holo complexes, which overlooks informative components like the filler (ligands and solvent) and suppresses intrinsic protein flexibility. To address this, the paper introduces EDMolGPT, a decoder-only autoregressive framework that generates 3D molecules conditioned on low-resolution electron density (ED) point clouds derived from the filler. By unifying calculated ED (CalED) for pre-training and experimental ED (ExpED) for fine-tuning, the method bridges computational scalability with physically grounded, realistic binding environments. Overall, the submission considers an important concept, offering a shift from rigid-pocket modeling to continuous, density-guided molecular generation.
Strengths and weaknesses

**Compliance With Llm Reviewing Policy:**

Affirmed.

**Final Justification:**

The rebuttal addressed my concern, I'll raise my score.

**Key Questions For Authors:**

1. While the authors position their work as an advance beyond traditional SBDD, all benchmarks are still conducted against standard SBDD baselines. Since the conditioning signal (filler ED) is directly derived from reference ligands, information leakage is unavoidable. The method behaves more like a high-performance ligand refinement or reconstruction tool rather than a genuine de novo design engine, making its true contribution hard to evaluate under the current SBDD evaluation framework.

2. The core motivation is to overcome the “rigid pocket assumption”. However, the main binding affinity metric relies on GlideSP, assessed through “min-in-place” and “redocking” protocols. Standard docking algorithms treat the receptor as rigid (or allow only highly restricted side-chain flexibility). Evaluating a “flexibility-aware” generative method using a rigid scoring function is logically inconsistent and contradictory.

3. Why does varying the resolution (d min​) lead to negligible performance differences in Table 3? Does this imply that the model merely learns spatial occupancy patterns instead of physically meaningful fine-grained features?

**Strengths And Weaknesses:**

Strengths:
1. The paper introduces a novel perspective by utilizing electron density (ED) from the "filler" (ligand and solvent) rather than relying solely on rigid, empty protein pockets. This bridges a gap between physical density maps and generative modeling.

2. The methodology elegantly unifies computational ED (CalED) for pre-training and experimental ED (ExpED) for fine-tuning within an autoregressive framework, adapting the FSMILES representation for 3D coordinates.

Weaknesses:
The core weaknesses center on conceptual ambiguity, evaluation inconsistency, insignificant gains and reproducibility barriers.
1. The method relies heavily on holo structures to extract filler ED, which substantially restricts its applicability in realistic de novo drug design where no known ligand is available.

2. The overall narrative and experimental design require better organization to clearly define and distinguish its actual task setting.

3. The advantages over SBDD baselines are not consistently strong across the adopted benchmarks. Moreover, the key differences between evaluations on DUD-E and CrossDock should be explicitly clarified, especially since the ranking of SBDD models on DUD-E differs notably from typical results.

4. The description of the original training dataset is insufficient to clarify its composition, and the data preprocessing steps need to be documented in greater detail for transparency and reproducibility.

---

> ### Author Rebuttal · Authors · 2026-03-31
>
> We sincerely appreciate your valuable feedback. We promise that we will add all contents to the original paper.
> >weakness1
>
> ***response:*** Please refer to our response to Reviewer 79Gc regarding the "Pharmacophore Dependence" concern.
>
> >weakness2
>
> ***response:***   We will carefully restructure the paper to better define the problem setting, more clearly distinguish training and inference scenarios, and further highlight the differences from prior SBDD paradigms. We will also refine the organization of the experimental section to ensure it is more closely aligned with the task definition and evaluation objectives.
> >weakness3
>
> ***response:*** We would like to clarify that the CrossDock dataset is constructed from artificially docked protein–ligand pairs, where ligands are placed based on geometric or shape complementarity and may not necessarily exhibit true binding activity. In contrast, DUD-E covers over 100 diverse protein targets, including key therapeutic classes such as Kinases, Proteases, GPCRs, and Ion Channels, with an average of over 200 active ligands per target. Additionally, each active compound in DUD-E is backed by experimentally measured affinities, providing a reliable basis for comparing generated molecules against verified bioactive ligands across a broad chemical space. This fundamental difference can lead to inconsistent model rankings across the two benchmarks.
> >weakness4
>
> ***response:***  We will give more detailed descriptions about electron density data in Sec. 3.2. We will also release the processed density files, generation code, and trained checkpoints, along with a full description of the preprocessing steps in the revised manuscript.
> >Question1
>
> ***response:*** Regarding the concern of information leakage and task positioning, we point out that our method is designed as a genuine de novo engine rather than a mere reconstruction tool. As reported in Tab.3, the structural similarity between the generated molecules and the reference ligands is consistently low, demonstrating the model's capacity to explore chemical space far beyond the reference binder. Furthermore, we provide an extensive discussion on information leakage in Appendix F.1. Additionally, a key distinction is that during the inference phase for ExpED, the model incorporates solvent-derived electron density, a modality that is excluded during training. This ensures that the model must generalize to unseen density distributions at test time rather than simply memorizing training references.
>
> Furthermore, as addressed in our response to Reviewer 79Gc regarding the "Pharmacophore Dependence", the operational scope and applicability of our method are fully consistent with current pocket-based SBDD frameworks. Therefore, evaluating our model against standard SBDD baselines is appropriate. As a pioneering work in utilizing electron density for structure-based molecular generation, our approach provides a novel perspective on structure-based design and establishes a solid foundation for future research in density-guided generative modeling.
> >Question2
>
> ***response:*** We agree that using GlideSP as the main binding affinity metric cannot fully demonstrate flexibility. Therefore, we instead focus on Bioactive Molecule Recovery. Specifically, we identify molecules generated by our method and ED2Mol that have a similarity greater than 0.5 to known bioactive compounds for a given target, perform Glide docking, and record the number of targets for which the Glide score exceeds -5 (i.e., relatively poor scores under the rigid-pocket assumption).
> ||Targets with Glide > -5|
> |---|---|
> |ED2Mol|0|
> |Ours|2|
>
> Many known bioactive molecules receive poor Glide scores due to the rigid-pocket assumption. Our method generates such molecules, yet they are known actives, indicating it explores flexibility beyond what rigid docking can capture. This experiment provides strong evidence that our approach can overcome the limitations of rigid assumptions.
> >Question3
>
> ***response:*** We attribute this observation primarily to our current representation scheme, which utilizes a fixed number of 199 sampling points to describe the "filler" electron density. Given the inherent noise associated with representing a complex 3D volume with such a limited number of points, this relatively coarse representation becomes the dominant factor (or bottleneck) in our current model, effectively diminishing the sensitivity to variations in $d_{min}$. However, this does not imply that the model merely learns simple spatial occupancy patterns.  We conduct additional ablation studies in Appendix F.4, where we change the number of sampling points and cancel specific labels for hydrogen-bond donors and acceptors. Those results show performance variations, confirming that the model is indeed capable of capturing physically meaningful fine-grained features when the representation capacity is increased.

---

> > ### Author Rebuttal · Reviewer_K9e2 · 2026-04-02
> >
> > I'll raise my score, thank you for your response.

---

> > > ### Author Response · Authors · 2026-04-02
> > >
> > > We sincerely appreciate your recognition of our work. Your insightful comments have helped us significantly, and we truly thank you for your time and effort.

---

### Official Review · Reviewer_Agry · 2026-03-10

**Soundness:** 3
**Presentation:** 3
**Significance:** 3
**Originality:** 4
**Overall Recommendation:** 5
**Confidence:** 3

**Summary:**

The authors address the gap that SBDD generative methods relying on empty binding pockets (holo) do not account for conformational flexibility. They propose conditioning ligand generation on electron density (ED) derived from the filler (solvent molecules & other ligands) instead, without modeling the target binding pocket. Given their method must model both the low-res filler ED and the molecule, they create a data pre-processing pipeline for generating point clouds in reciprocal space and assigning pharmacophores, featurizing the filler ED. To featurize the molecule, they propose an improvement to FSMILES that avoids fragment splitting that’s used in conjunction with discretized x, y, z coordinates relative to the molecule’s geometric center and relative bond angles and dihedral angles between each atom and the three atoms next-closest atoms to the center. This discretization allows them to use an autoregressive model EDMolGPT that generates atoms ordered by x, y, z coordinates relative to the molecule center. They evaluate EDMolGPT against other pocket-conditioned generative models on bioactive molecular recovery based on DUD-E, binding affinity as measured with GlideSP, conformational stability with PoseCheck, and molecular properties such as QED, SAS & average weight. They further evaluate EdMolGPT on experimentally-resolved EDs and carry out ablations of the diffraction resolution (coarseness of ED), number of sampling points (sparsity) and the sampling temperature T.

**Compliance With Llm Reviewing Policy:**

Affirmed.

**Final Justification:**

The use of filler electron density for conditioning ligand generation is an original and well-motivated approach to modeling target flexibility, and the methodology, experiments, and ablations are thorough. The rebuttal satisfactorily addressed my concerns, thus I am maintaining my score of 5

**Key Questions For Authors:**

- Could you clarify the audience and purpose of your current discussion in the main text? Please consider replacing this with the current discussion mentioned in the Appendix
- Could you explain in greater detail why you chose an autoregressive model? In the intro, you highlight its simplicity, flexibility and high efficiency, but don’t address its potential drawbacks of “Discretizing out” information. It’s clear that you choose the filler ED to overcome the limitations of rigid pockets, but why would an autoregressive model be the right modeling choice in this setting?

**Limitations:**

yes

**Strengths And Weaknesses:**

## Soundness:
Strengths:
- The authors comprehensively explain their methodology, experiments, and support the claims and motivations for filler ED-conditioned generation. Experiments are thorough, and the authors fairly evaluate their method with baselines, flagging differences that cannot be controlled for (e.g. ED2Mol applying an extra refinement step)

Weaknesses:
- The authors highlight weaknesses of the method and provide mostly satisfying explanations when the model is underperforming baselines, though on conformational stability, I’d argue the EDMolGPT strain energies are worse than Lingo3DMol (not on par). And then for min-in-place binding affinity, this metric is valued in the CalED setting when EDMolGPT does best, but then de-valued in the ExpED setting
- As a minor point, positional embeddings might not be needed in EDMolGPT as the token embeddings capture discrete distances between atoms. It might be helpful to have one line explaining whether (and why) position encodings are needed in this case
- In 3.4, you mention you employ linear classifiers “since all items are converted into discrete space”. Could you expound on what you mean by this? E.g. do the features need to be in discrete space to use linear classifiers? And why do you need them for predicting a, v, l, theta, etc.?
- DUD-E is quite small for assessing Biomactive Molecule Recovery, which is the metric you lead with. Does it represent a broad enough set of bioactive molecules to conclude that similarity with its molecules are similar to all known active compounds? Consider addressing the limitations of such a small reference set.
- To verify the hypothesis that baselines might achieve favorable QED/SAS scores b/c they generate smaller, simpler molecules, I’d suggest binning samples by weight and computing drug-like properties for each bin.


## Presentation:
Strengths:
- The intro through methodology is well-articulated and explained. The evaluations and experiments  are explained at a sufficient level of detail detail, including analysis
- The appendix is well-structured and organized

Weaknesses:
- The Discussion (section 5) in the main body does not have a clear purpose or audience. It’s also unclear whether you’re trying to highlight the differences in methodology that sets EDMolGPT apart or if you’re concerned others won’t see your method as consistent with other SBDD methods that use the holo structure. Are you drawing constraints or ensuring consistency with other SBDD methods? I’d consider removing this entirely. If there’s still space after revision, I’d advise bringing the Discussion that’s currently in the appendix into the main body, replacing the current discussion. The discussion in the appendix is more comprehensive and better compares EDMolGPT with others SBDD methods at a high level.
- I’d recommend giving Figure 3, 5, and 6 their own figure panel instead of crowding them, as you’re likely planning on for a camera-ready version
- Considering referencing section C.1 in 3.3 when you describe that you improve FSMILES

## Signfiicance:
Strengths:
- Their method indeed is significant for its focus on including the filler’s ED into conditional generation, ignoring rigid pockets entirely, which has not previously been explored. It tackles the reality that rigid holo structures do not model protein flexibility and conformational changes upon binding
- It’s likely that others will build on using filler EDs for ligand generation, and they’ve set solid groundwork for others to build off
- They make their data available and suggest they’ll make their code available on release, which would be very welcome for adoption and reproducibility

Weaknesses:
- Others may hesitate to use EdMolGPT directly as it requires discretizing coordinates and artificially ordering tokens for autoregressive generation. This, along with low resolution filler EDs may, in theory, ablate too much continuous information and prevent wider adoption
- As a minor point, you could explicitly call out that your featurization and EDMolGPT are both invariant to rotations and translations, despite the discretization of 3D coordinates

## Originality:
Strengths:
- The authors explore the intriguing problem of using filler EDs for conditional generation instead of fixed binding pockets. This alone is underexplored.
- They also further explore the use of a modeling method that requires discretizing a continuous space, and show its competitiveness with continuous methods.
- They also make available their curated data for 2 million pre-processed molecules
- The contributions are indeed distinguished from existing pocket-conditioned ligand generative methods

---

> ### Author Rebuttal · Authors · 2026-03-31
>
> We sincerely appreciate your valuable comment. We promise that we will add all contents to the paper.
> >Sound. weakness1
>
> ***response:*** We agree that the SE of EDMolGPT is higher than that of Lingo3DMol, though still within a comparable range, indicating reasonable but not optimal conformational stability. The differing Min-in-place trends between CalED and ExpED mainly arise from: (1) limited high-quality experimental Holo structures, which constrain ExpED training despite CalED pretraining; and (2) the higher flexibility and noise in experimental electron density, making evaluation less stable. Overall, this work is an initial exploration of leveraging electron density for molecule generation, and we acknowledge that some metrics do not yet surpass strong baselines. We will continue improving stability in future work.
> >Sound. weakness2
>
> ***response:*** Token embeddings capture discretized inter-atomic distances, but positional encodings are still needed to preserve ordering and provide a global spatial reference. They also help distinguish electron-density and molecular tokens, improving spatial awareness during decoding.
> >Sound. weakness3
>
> ***response:*** Our intention was to highlight the distinction between prediction paradigms for continuous and discrete spaces. Continuous variables are typically handled via regression (a single linear layer), whereas discretization reformulates the task as classification over bins, requiring a linear classifier (linear layer + softmax). Here, “linear classifier” refers to the discretized objective rather than any restriction on input features. We will revise the manuscript to clarify this point and avoid ambiguity.
> >Sound. weakness4
>
> ***response:*** We evaluate our model on DUD-E, a widely adopted SBDD benchmark with two key advantages. First, DUD-E provides broad coverage of protein target types, comprising over 100 diverse targets across major therapeutic classes, including kinases, proteases, GPCRs, and ion channels. This diversity enables assessment across a wide range of binding environments and biological functions. Second, all active compounds in DUD-E are supported by experimentally measured affinities, ensuring that the reference bioactive molecules are biologically validated, making DUD-E a well-balanced benchmark for evaluating bioactive molecule recovery. We acknowledge that no single dataset can capture the full chemical space of all known active compounds while maintaining broad coverage across diverse target types. Therefore, similarity to DUD-E actives should not be interpreted as similarity to all possible bioactive molecules, but rather as a consistent proxy widely used for comparative evaluation. Our evaluation follows established protocols in generative modeling, ensuring fair comparison with other methods. We agree that extending evaluation to larger and more diverse datasets would be a valuable direction for future work.
> >Sound. weakness5
>
> ***response:***  We have summarized results for ED2Mol and EdMolGPT in the table below:
> |Molecular Weight|SAS(Ed2Mol/Ours)|QED(Ed2Mol/Ours)|
> |---|---|---|
> |<180|**3.18**/3.29|**0.66**/0.52|
> |180-300|3.93/**3.80**|**0.75**/0.63|
> |300-420|4.68/**3.73**|**0.74**/0.63|
> |>420|5.27/**3.88**|**0.52**/0.46|
>
> The binned analysis shows that while Ed2Mol achieves higher QED in certain ranges, its SAS increases significantly with molecular weight, indicating reduced synthetic accessibility for larger molecules. In contrast, EdMolGPT maintains more stable and favorable SAS across all weight bins. Moreover, considering metrics such as recovery rate and binding affinity, EdMolGPT demonstrates stronger overall performance in generating biologically relevant structures. We acknowledge that Ed2Mol has strengths in optimizing specific drug-likeness properties, and view it as a valuable benchmark. In future work, we plan to incorporate these advantages to further improve the chemical quality of our generated molecules.
> >Present. weakness.
>
> ***response:***  We will revise the manuscript according to your advice. Thank you for your detailed advice.
> >Significance weakness1
>
> ***response:*** We adopt an autoregressive formulation, as prior work (e.g., Lingo3DMol) shows it often improves drug-likeness and chemical validity.  Given that molecular components, such as element types, bond orders, and their combinatorial rules, are intrinsically discrete, they form a structured "grammar" that sequential modeling is uniquely equipped to capture. While discretization trades off some continuous information, it serves as a first step toward leveraging transformers for ED-based drug design.
> >Significance weakness2
>
> ***response:***  We will include an explanation in the revised manuscript.
> >Questions1
>
> ***response:*** This discussion clarifies our new paradigm for researchers across disciplines. We will merge the Appendix into the main text to enhance conciseness as suggested.
> >Questions2
>
> ***response:*** See response to significance weakness1.

---

> > ### Author Rebuttal · Reviewer_Agry · 2026-04-02
> >
> > Thank you for your detailed responses and analysis. I will maintain my score of 5, with the following feedback:
> >
> > **Molecular Weight Binned Analysis (QED and SAS)**  The data supports the claim regarding synthetic accessibility: EDMolGPT maintains a stable SAS across weight bins, whereas ED2Mol’s SAS degrades for heavier molecules. However, the data does not support the implication that EDMolGPT achieves comparable QED. ED2Mol's QED scores are higher across every weight bin, and both models follow the same curve. You might considering calling this out in the manuscript as a tradeoff: EDMolGPT provides synthetic stability for larger molecules at the cost of lower overall QED.
> >
> > **Architectural Clarifications** The architectural explanations are logical. Clarifying the "linear classifier" terminology (discretized objective vs. continuous inputs) resolves the ambiguity. The rationale for retaining positional encodings to distinguish between point cloud and molecular tokens is also sound. Please ensure these clarifications are explicitly integrated into the revised text.
> >
> > **Baselines and Datasets** Acknowledging the DUD-E dataset's limitations as a "consistent proxy" is appropriate. The explanation regarding conformational stability on ExpED (due to inherent noise and data scarcity) is acceptable as a baseline for future work.
> >
> > **Conclusion** Moving the broader SBDD discussion from the appendix to the main text will better contextualize the method. The responses are satisfactory, and using filler ED for conditioning remains an original approach to modeling target flexibility.

---

> > > ### Author Response · Authors · 2026-04-03
> > >
> > > Thank you for your positive evaluation and for the constructive comments. We sincerely appreciate the time and effort you have invested in reviewing our work. Your insightful feedback is highly valuable and has been very helpful in improving both the clarity and quality of our paper. We will carefully incorporate your suggestions in the revision.

---

### Official Review · Reviewer_79Gc · 2026-03-13

**Soundness:** 3
**Presentation:** 3
**Significance:** 3
**Originality:** 3
**Overall Recommendation:** 4
**Confidence:** 3

**Summary:**

This paper introduces EDMolGPT, a novel decoder-only autoregressive framework for 3D structure-based drug design (SBDD). Unlike conventional methods that rely on rigid geometric abstractions of empty protein pockets, this approach utilizes low-resolution electron density (ED) derived from the "filler" (ligands and solvent molecules) as a physically grounded conditioning signal. The model represents molecules using an improved Fragment SMILES (FSMILES) and discretized 3D coordinates, enabling end-to-end generation of novel ligands compatible with dynamic binding environments.

**Compliance With Llm Reviewing Policy:**

Affirmed.

**Key Questions For Authors:**

NA

**Limitations:**

Pharmacophore Dependence: The model relies on assigning pharmacophore labels to the point cloud based on the nearest atom in the filler. This creates a high dependency on chemical priors that might not be available in strictly de novo scenarios where a high-quality reference binder is absent.

Resolution and Sparsity Trade-offs: While ablation studies suggest that the number of points ($N_p = 199$) is critical, the choice appears somewhat empirical. Further discussion on how this sparsity affects performance across significantly different pocket volumes would be beneficial.

Domain Gap in Intensity: There is a noticeable discrepancy in intensity distributions between CalED and ExpED due to experimental noise and flexibility. The manuscript would benefit from a deeper theoretical analysis of how the fine-tuning stage mitigates this distributional shift.

**Strengths And Weaknesses:**

Physically Grounded Representation: Using filler ED instead of static pocket geometry effectively captures ensemble-averaged spatial distributions and interaction patterns, allowing the model to account for protein flexibility and conformational adaptations.

Architectural Innovation: As the first decoder-only approach in this domain, EDMolGPT combines the high capacity and efficiency of GPT-style models with 3D structural reasoning.

Unified Training Strategy: The two-stage training process—large-scale pre-training on calculated density (CalED) followed by fine-tuning on experimental density (ExpED)—bridges the gap between abundant computational data and scarce experimental measurements.

Superior Performance: Evaluations on 101 DUD-E targets demonstrate that EDMolGPT achieves a higher bioactive molecule recovery rate (41%) and better binding modes compared to established baselines like Pocket2Mol and TargetDiff.

---

> ### Author Rebuttal · Authors · 2026-03-31
>
> We sincerely appreciate the insightful feedback. In the following, your questions are responsed point-by-point. We promise that we will add all contents to the original paper.
>
> >Pharmacophore Dependence.
>
> ***response:*** Regarding the pharmacophore dependence, we clarify that *de novo* design does not strictly equate to using ligand-free *Apo* structures. In practice, the majority of successful de novo cases are initiated from empty pockets derived from *Holo* structures (e.g., proteins crystallized with endogenous ligands like ADP or NADPH, which are subsequently removed) [c]. This workflow is preferred because *Apo* and *Holo* pockets exhibit significant conformational discrepancies in volume, surface area, and hydrophobicity [a, b]. Our method leverages these *Holo*-like cavities where natural endogenous substrates inherently provide essential pharmacophoric information. Even if the endogenous substrate's structure is relatively simple, we can still extract pocket-derived pharmacophore features and map them onto the filler density grids [e], ensuring robust guidance even in the absence of a high-quality reference binder.
>
> For scenarios where only an *Apo* structure is available, current state-of-the-art strategies typically follow the logic of reconstructing a *Holo*-state prior to molecule design [d]. For instance, one can utilize homologous templates to infer the *Holo*-conformation and place fillers to represent the binding environment. Following molecular dynamics relaxation, our framework can seamlessly utilize the resulting calED and corresponding pharmacophoric priors to guide the generation process. Therefore, our reliance on filler-informed density does not restrict applicability; rather, it mirrors the rigorous biophysical preparations, such as identifying conserved interaction hotspots, required for successful computational discovery in realistic drug design pipelines.
>
> [a] Guo, Zuojun, et al. Identification of protein–ligand binding sites by the level-set variational implicit-solvent approach. Journal of Chemical Theory and Computation 11.2 (2015): 753-765.
>
> [b] Clark, Jordan J., et al. Inherent versus induced protein flexibility: Comparisons within and between apo and holo structures. PLoS computational biology 15.1 (2019): e1006705.
>
> [c] Lam, Jordy Homing, and Vsevolod Katritch. Navigating structure-based drug discovery with emerging innovations in physics-and knowledge-based approaches. npj Drug Discovery 2.1 (2025): 29.
>
> [d] Zhang, Jinze, et al. Holo protein conformation generation from apo structures by ligand binding site refinement. Journal of Chemical Information and Modeling 62.22 (2022): 5806-5820.
>
> [e] Flynn, Emma L., et al. PharmacoForge: pharmacophore generation with diffusion models. Frontiers in Bioinformatics 5 (2025): 1628800.
>
> >Resolution and Sparsity Trade-offs.
>
> ***response:*** We conduct ablations on $N_p$ in Tab.6 in the paper. Increasing $N_p$ yields a finer representation of density patterns, improving Min-in-place scores but slightly reducing diversity, while also increasing the number of input tokens and thus computational cost. In contrast, using fewer points reduces token count and resource consumption but leads to some performance degradation. Therefore, $N_p=199$ represents a practical balance between resource efficiency and performance.
>
> >Domain Gap in Intensity.
>
> ***response:*** We provide a preliminary theoretical perspective based on the generalization bound. Let $D_{cal}$ and $D_{exp}$ denote the distributions of calculated and experimental electron densities, respectively. For a model $G_{\theta}$ pre-trained on $D_{cal}$, its risk on the experimental domain, $\epsilon_{exp}(G_{\theta})$, can be bounded as follow [f]:
>
> $\epsilon_{exp}(G_{\theta}) \leq \epsilon_{cal}(G_{\theta}) + \frac{1}{2} d_{\mathcal{H}\Delta\mathcal{H}}(D_{cal}, D_{exp}) + \lambda,$
>
> where $d_{\mathcal{H}\Delta\mathcal{H}}$ measures the divergence between the two distributions, and $\lambda$ is a constant related to the complexity of the hypothesis class. Due to experimental noise and molecular flexibility, the initial divergence $d_{\mathcal{H}\Delta\mathcal{H}}(D_{cal}, D_{exp})$ is non-negligible, which may lead to a relatively loose upper bound on $\epsilon_{exp}(G_{\theta})$.
>
> Fine-tuning on $D_{exp}$ can be viewed as adapting the model toward the target distribution, which helps reduce the effective distribution discrepancy to some extent. In an idealized setting, this process would correspond to replacing the divergence term with $d_{\mathcal{H}\Delta\mathcal{H}}(D_{exp}, D_{exp}) = 0$. In practice, while such a discrepancy cannot be entirely eliminated, it can be substantially mitigated through fine-tuning, thereby tightening the generalization bound and improving performance on experimental data.
>
> [f] Ben-David, Shai, et al. Analysis of representations for domain adaptation. Advances in neural information processing systems 19 (2006).

---

> > ### Author Rebuttal · Reviewer_79Gc · 2026-04-03
> >
> > Thanks for your explainations!

---

> > > ### Author Response · Authors · 2026-04-03
> > >
> > > Thank you for your positive evaluation and helpful comments. We greatly appreciate your feedback, which has been valuable for improving our work.

---

### Decision · Program_Chairs · 2026-04-30

**Decision:**

Accept (regular)

**Comment:**

Reviewers are generally positive about the goal of moving beyond rigid pocket geometry, and agree that the use of electron density from the filler, agreeing that it is underexplored, and that the paper makes a useful contribution in addressing this problem.

At the same time, reviewers draw attention to the question as to whether (e.g. hbu3) "the model learns general binding principles versus relying on the reference ligand's geometry" or (k9e2) "information leakage is unavoidable".
In rebuttal, the authors observe that "the generated molecules have low structural similarity to reference ligands", "demonstrating exploration beyond known binders".  However, low structural similarity might occur for several reasons, so it is not a complete rebuttal of the reviewers' objections.